# STAT3 in tumor fibroblasts promotes an immunosuppressive microenvironment in pancreatic cancer

Julia E Lefler[1], Catherine B MarElia-Bennett[1], Katie A Thies[1], Blake E Hildreth III[1], Sudarshana M Sharma[1], Jason R Pitarresi[2], Lu Han[1], Caroline Everett[1], Christopher Koivisto[1], Maria C Cuitino[1], Cynthia D Timmers[1], Elizabeth O'Quinn[1], Melodie Parrish[1], Martin J Romeo[1], Amanda J Linke[1], G Aaron Hobbs[1], Gustavo Leone[3], Denis C Guttridge[4], Teresa A Zimmers[5], Gregory B Lesinski[6], Michael C Ostrowski[1]

**Pancreatic ductal adenocarcinoma (PDAC) is associated with an incredibly dense stroma, which contributes to its recalcitrance to therapy. Cancer-associated fibroblasts (CAFs) are one of the most abundant cell types within the PDAC stroma and have context-dependent regulation of tumor progression in the tumor microenvironment (TME). Therefore, understanding tumor-promoting pathways in CAFs is essential for developing better stromal targeting therapies. Here, we show that disruption of the STAT3 signaling axis via genetic ablation of *Stat3* in stromal fibroblasts in a *Kras*[G12D] PDAC mouse model not only slows tumor progression and increases survival, but re-shapes the characteristic immune-suppressive TME by decreasing M2 macrophages (F480+CD206+) and increasing CD8+ T cells. Mechanistically, we show that loss of the tumor suppressor PTEN in pancreatic CAFs leads to an increase in STAT3 phosphorylation. In addition, increased STAT3 phosphorylation in pancreatic CAFs promotes secretion of CXCL1. Inhibition of CXCL1 signaling inhibits M2 polarization in vitro. The results provide a potential mechanism by which CAFs promote an immune-suppressive TME and promote tumor progression in a spontaneous model of PDAC.**

## Introduction

One of the most prominent histopathological attributes of pancreatic ductal adenocarcinoma (PDAC) is a desmoplastic stroma that develops stepwise along with the tumor. Remarkably, malignant cells account for less than 20% of the total tumor volume in primary PDAC tumors, whereas the stroma can account for more than 70% (Yang et al, 2012). Within this stroma, cancer-associated fibroblasts (CAFs) are one of the most abundant cell populations and can comprise 15–85% of the tumor mass (Lafaro & Melstrom, 2019; Liu et al, 2019). CAFs can contribute to carcinogenesis and tumor progression, but previous studies have suggested the activities of CAFs within the tumor microenvironment (TME) are much more nuanced (Amakye et al, 2013; Ozdemir et al, 2014; Rhim et al, 2014). Furthermore, identification of subtypes of CAFs within the PDAC TME with both tumor-promoting and tumor-limiting characteristics have also complicated our understanding of CAF contribution to PDAC pathology (Sherman et al, 2014; Ohlund et al, 2017; Biffi et al, 2019; Elyada et al, 2019; Dominguez et al, 2020). Overall, these studies highlight the need for a more detailed examination of the important signaling pathways that are active between CAFs, tumor cells, and other stromal cell compartments.

Loss of the tumor suppressor phosphatase and tensin homolog (PTEN) is commonly observed in solid cancers and is associated with more aggressive tumors (Liaw et al, 1997; Nowak et al, 2015; Chen et al, 2018). Our work has also uncovered non–cell-autonomous functions of stromal PTEN in suppressing tumor growth in breast cancer (Trimboli et al, 2009; Bronisz et al, 2011; Sizemore et al, 2018). Similarly, our group revealed a correlation between PTEN loss in CAFs and reduced overall survival of PDAC patients, which was confirmed experimentally in mouse models (Pitarresi et al, 2018). Furthermore, PTEN deletion has been linked to evasion of the immune response in several tumor types (Vidotto et al, 2020). Genetic studies from our group have also shown that depletion of PTEN in the stroma increased the infiltration of macrophages in both breast and pancreatic cancer mouse models (Trimboli et al, 2009; Liu et al, 2016). To address these observations,

[1]Hollings Cancer Center and Department of Biochemistry and Molecular Biology, Medical University of South Carolina, Charleston, SC, USA   [2]Division of Gastroenterology, Department of Medicine and Abramson Cancer Center, University of Pennsylvania, Philadelphia, PA, USA   [3]Department of Biochemistry, Medical College of Wisconsin, Milwaukee, WI, USA   [4]Department of Pediatrics and Hollings Cancer Center, Medical University of South Carolina, Charleston, SC, USA   [5]Department of Anatomy, Cell Biology, and Physiology, Indiana University School of Medicine, Indianapolis, IN, USA   [6]Department of Hematology and Medical Oncology, Winship Cancer Institute of Emory University, Atlanta, GA, USA

Correspondence: ostrowsk@musc.edu

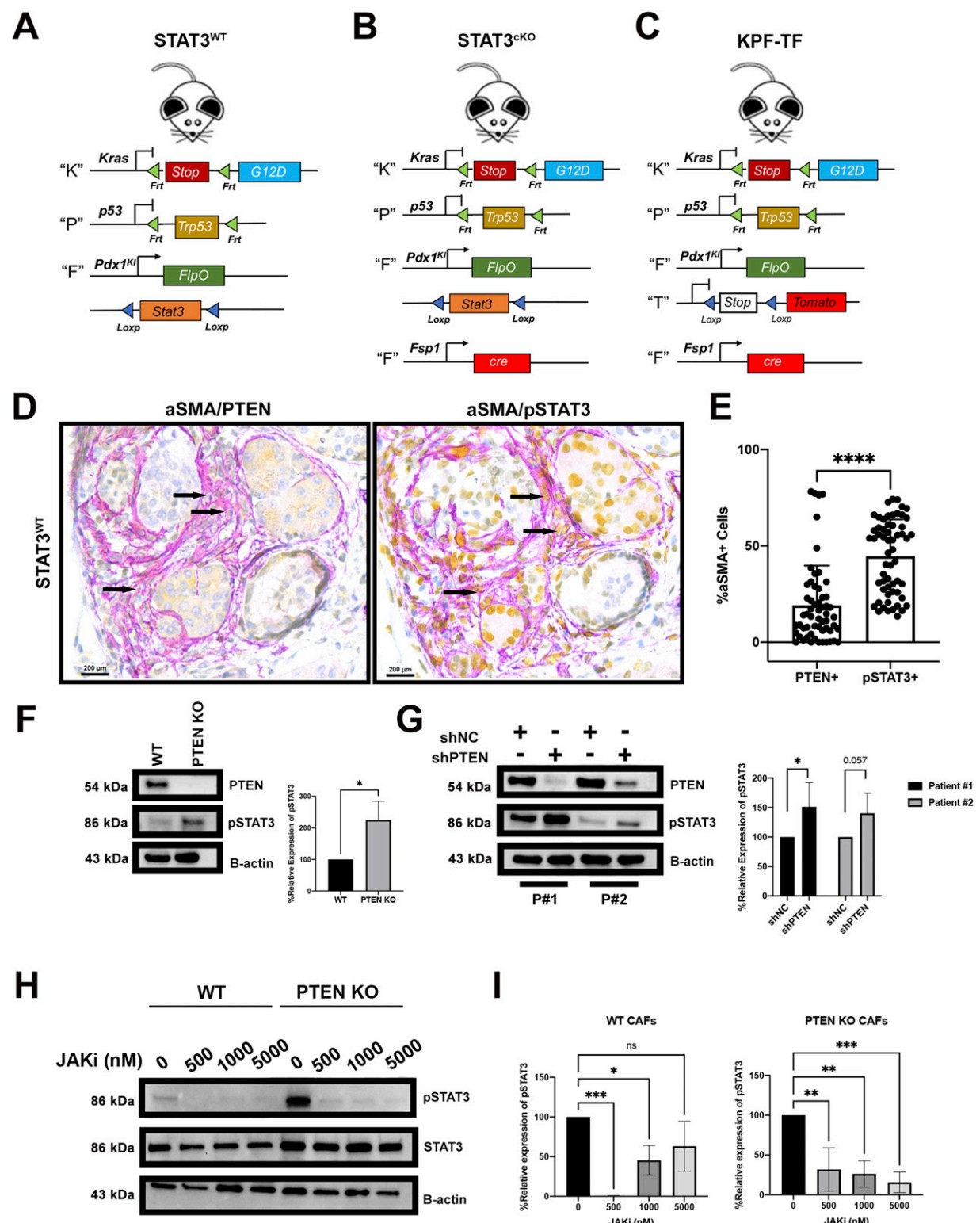

**Figure 1. Loss of PTEN in stromal PDAC Fibroblasts results in activation of STAT3.**

**(A)** Schematic of genotype of the STAT3[WT] mouse. **(B)** Schematic of genotype of the STAT3[cKO] mouse. **(C)** Schematic of genotype of the KPF-TF mouse. **(D)** Immunohistochemical staining of pancreas lesion in STAT3[WT] mouse. Panel on left displays dual PTEN (roche yellow) and aSMA (roche purple), whereas panel on right displays dual pSTAT3-Tyr705 (roche yellow) and aSMA (roche purple). **(E)** Quantification of PTEN/aSMA and pSTAT3/aSMA double-positive cells in the STAT3[WT] lesions (n = 5). **(F)** Western blot analysis for PTEN, pSTAT3-Tyr705, and B-ACTIN in PTEN WT and PTEN KO mouse pancreatic fibroblasts and quantification of WB in using densitometry analysis. **(G)** Western blot analysis for PTEN, pSTAT3-Tyr705, and B-ACTIN in shPTEN or shNC human pancreatic cancer fibroblasts and quantification of WB in

we aimed to identify important downstream effectors of PTEN loss that promote tumor growth and an immune-suppressive TME in PDAC.

STAT3 is a transcription factor downstream of gp130-like receptors that activates expression of genes involved in differentiation, development, proliferation, and apoptosis (Schindler et al, 2007; Stark & Darnell, 2012). STAT3 plays an important role in solid tumors, including in KRAS-induced pancreatic tumorigenesis (Corcoran et al, 2011). IL-6 is an important canonical activator of STAT3 and the IL-6/STAT3 axis promotes the expansion of immunosuppressive cells while altering the balance of T-cell subsets (Mace et al, 2018; Korn & Hiltensperger, 2021). Notably, IL-6 and activated pY705-STAT3 are evident in CAFs in human PDAC samples and isolated CAFs (Mace et al, 2018); however, the role of STAT3 signaling from CAFs within the context of de novo pancreatic tumorigenesis is unknown. Thus, we hypothesized that PTEN modulation of activated STAT3 in CAFs results in downstream changes in the immune microenvironment in PDAC.

In the present work we show that PTEN is a negative regulator of STAT3 activation in PDAC CAFs. In addition, STAT3 signaling from CAFs promotes tumorigenesis and drives an immunosuppressive microenvironment in PDAC tumors.

# Results

## Loss of PTEN in pancreatic CAFs results in STAT3 activation

Whether loss of PTEN expression results in an up-regulation of STAT3 activation in pancreatic CAFs is unknown. To understand this relationship in vivo, we used the "KPF" (*FSF-Kras$^{G12D/+}$*; *Pdx1FlpO$^{ki/+}$*; *Trp53$^{frt/+}$*) knock-in PDAC model developed by our group (Wu et al, 2017) (Fig 1A). This FlpO-based model allows simultaneous genetic manipulation of stromal cells using Cre recombinase. We then performed dual IHC for aSMA (purple)/PTEN (yellow) on pancreas sections from 90-d-old KPF mice, a time when all the mice have PanIN lesions (Fig 1D). We quantified the colocalization of the two markers (appearing orange) using multispectral imaging (see the Materials and Methods section) (Fig 1E). Remarkably, this analysis demonstrated that PTEN protein expression was absent in ~80% of aSMA+ cells located in the tumor microenvironment (Fig 1E). In contrast, PTEN expression was robust in the tumor epithelial cells (Fig 1D). In parallel, consecutive sections were stained with aSMA (purple)/pY705-STAT3 (yellow), revealing 45% of aSMA+ fibroblasts were also pY705-STAT3+ (Fig 1E). We observed a similar absence of PTEN expression in the stroma in the KPC mouse model (Fig S1A). To further confirm the relationship between PTEN and activated STAT3, we assessed whether STAT3 phosphorylation was increased in PTEN-null fibroblasts (Pitarresi et al, 2018). Western blot analysis revealed that STAT3 phosphorylation at Tyr705 was increased in PTEN KO murine pancreatic CAFs (Fig 1F) (total STAT3 blot shown in Fig S1B). To determine whether STAT3 activation increased in response to PTEN loss in human CAFs, we knocked down PTEN from

two primary PDAC patient-derived CAF cell lines using shPTEN (Dharmacon). Western blot analysis confirmed that, like mouse CAF cell lines, STAT3 phosphorylation is indeed increased when PTEN is depleted in one of the human CAF lines and a trending increase in the other (Fig 1G) (total STAT3 blot shown in Fig S1D).

How PTEN negatively regulates pSTAT3 in pancreatic CAFs is unknown. We hypothesized the increase in STAT3 phosphorylation in the PTEN-null CAFs could be from canonical JAK signaling or noncanonical PI3k signaling. To investigate the possibility of the JAK pathway, we treated WT and PTEN-null mouse CAFs with a JAK1 inhibitor (Momelotinib). Western blot analysis revealed that upon treatment of the inhibitor, STAT3 phosphorylation significantly decreased in both WT and PTEN-null serum stimulated CAFs (Fig 1H and I). We then treated WT and PTEN-null CAFs with a pan PI3K inhibitor, and isoform specific inhibitors (PI3K-α and PI3K-γ) to investigate the contribution of the PI3K pathway to STAT3 activation in the PTEN-null CAFs. Interestingly, treatment of these inhibitors did not have a significant effect on STAT3 phosphorylation (Fig S1F).

## STAT3 signaling in PDAC CAFs promotes tumorigenesis

To study the role of fibroblast derived STAT3 on pancreatic tumorigenesis, we created a genetically engineered dual recombinase mouse model that combined the KPF model with fibroblast-specific Cre (*Fsp-Cre*)–mediated conditional knockout of *Stat3$^{LoxP}$* (Trimboli et al, 2008, 2009; Wu et al, 2017). Of note, the *Fsp-Cre* transgene developed by our laboratory does not recapitulate the expression pattern of endogenous *FSP1/S100A4* gene, probably because of the integration site of the transgene (Trimboli et al, 2008, 2009). This approach allows for activation of KRAS and loss of the tumor suppressor p53 in the pancreatic epithelium, as well as the deletion of STAT3 in CAFs. We were thus able to evaluate changes in the epithelium as well as the tumor microenvironment over the course of de novo tumor formation and progression in response to the conditional knockout of STAT3 in CAFs. This strategy was achieved by generating mice of two genetic groups (Fig 1A and B): control (denoted as STAT3$^{WT}$: *FSF-Kras$^{G12D/+}$*; *Pdx1FlpO$^{ki/+}$*; *Trp53$^{frt/+}$*; *Stat3$^{fl/fl}$*) and experimental (denoted as STAT3$^{cKO}$: *FSF-Kras$^{G12D/+}$*; *Pdx1FlpO$^{ki/+}$*; *Trp53$^{frt/+}$*; *Fsp-Cre*; *Stat3$^{fl/fl}$*).

Dual color immunofluorescence (IF) for aSMA and pSTAT3-Tyr705 on paraffin embedded tissue sections was used to quantify the extent of STAT3 deletion in the stroma of STAT3$^{WT}$ versus STAT3$^{cKO}$ (Figs 2A and S2A). This analysis revealed that ~38% of aSMA+ fibroblasts are pSTAT3+ within the STAT3$^{WT}$ TME, and the addition of the Fsp-Cre transgene in STAT3$^{cKO}$ model resulted in a significant twofold decrease in pSTAT3+aSMA+ fibroblasts in the STAT3$^{cKO}$ cohorts compared with control (Fig 2B). A similar analysis was performed using a distinct activated fibroblast marker, PDGFRB demonstrating a significant reduction in pSTAT3+Pdgfrb+ cells within the stroma of the STAT3$^{cKO}$ model (Fig S2B and C).

To evaluate the extent to which our *Fsp-Cre* targeted fibroblasts in our KPF model, we imaged pancreatic tumor tissue from a KPF mouse model that has an additional tomato reporter under the

using densitometry analysis. **(H)** Western blot analysis for pSTAT3, STAT3, and B-ACTIN in WT and PTEN KO mouse pancreatic fibroblasts treated with JAKi inhibitor. **(H, I)** Quantification of WB in (H) using densitometry analysis. Bars 100 μM. (*)P < 0.05 (****)P < 0.0001.

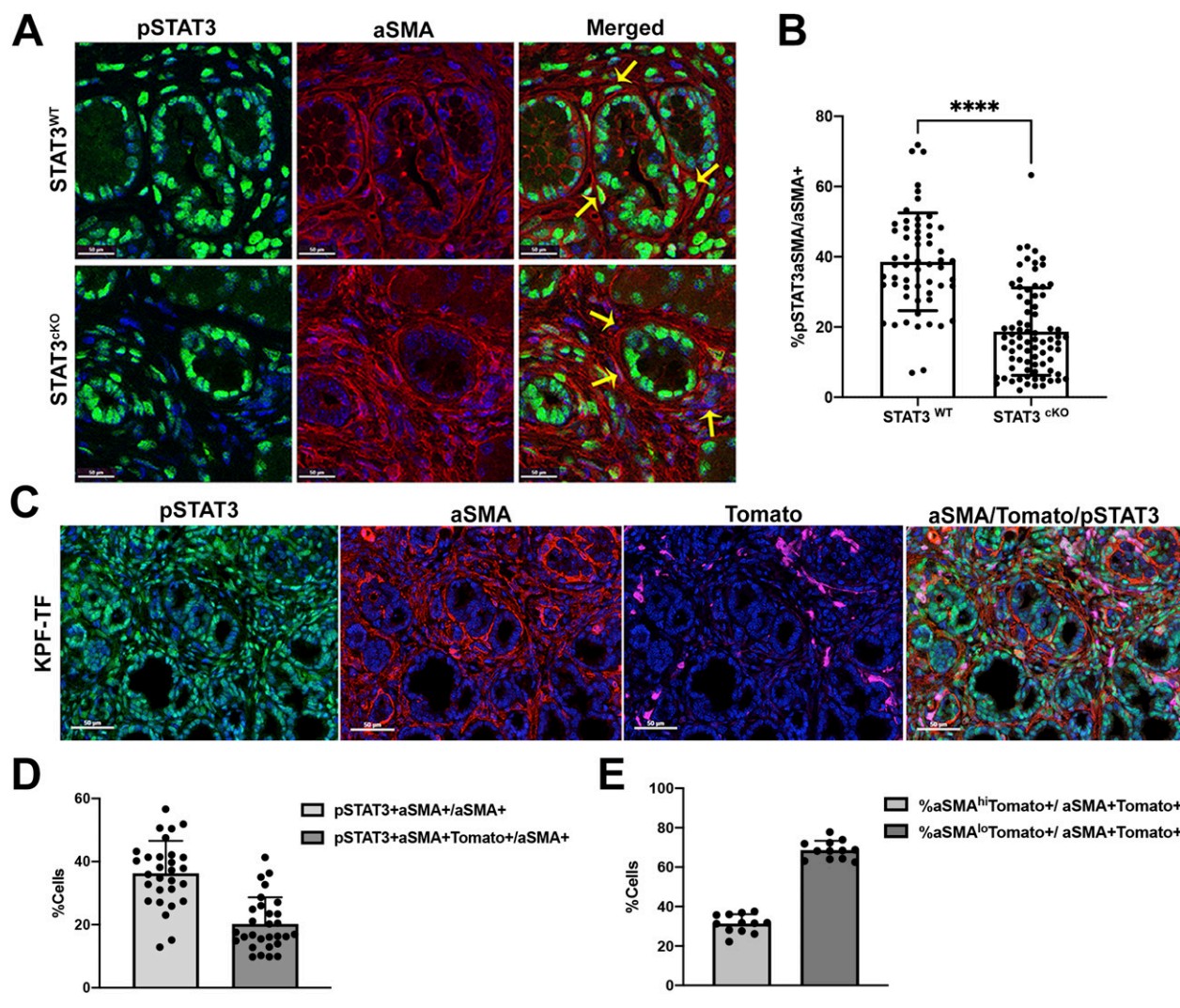

**Figure 2. Fsp-Cre targets aSMA+ fibroblasts in PDAC GEMMs.**
**(A, B)** Dual-color (IF) of pSTAT3-Tyr705 (green) and aSMA (red) expression and (B) quantification of % pSTAT3+aSMA+ fibroblasts out of total aSMA+ fibroblasts between both genetic groups (STAT3$^{WT}$: n = 5, STAT3$^{cKO}$: n = 6). **(C)** Multi-color IF of pSTAT3-Tyr705 (green), aSMA (red), and tomato (pink) expression in KPF-TF tumors. **(D)** Quantification of IF in KPF-TF tumors (n = 4). **(E)** Quantification of aSMA$^{Hi}$/Tomato+ Cancer-associated fibroblasts versus aSMA$^{Lo}$/Tomato+ Cancer-associated fibroblasts out of total aSMA+Tomato+ double-positive cells. Error bars represent mean ± SD. Bars 50 μM. (****)P < 0.0001.

control of a lox-stop-lox (LSL) cassette. This KPF mouse combined with *Fsp-Cre*, denoted as "KPF-TF" (*FSF-Kras$^{G12D/+}$; Pdx1FlpO$^{ki/+}$; Trp53$^{frt/+}$; Fsp-Cre; LSL-TDTomato$^{fl/fl}$*), allowed us to visualize the population of cells targeted by *Fsp-Cre* during tumorigenesis (Fig 1C). Using a three-color multiplex immunofluorescent stain for TdTomato, pSTAT3-Tyr705, and aSMA, we quantified the percentage of CAFs targeted in this model (Fig 2C). The Vectra multispectral platform was used to quantify the percentage of pSTAT3+aSMA+ fibroblasts and Fsp-Cre-TDtomato+pSTAT3+aSMA+ cells within the PDAC TME. The analysis confirmed that similar to the KPF model, ~36% of aSMA+ were pSTAT3+ in KPF-TF mice (Fig 2D). In addition, the analysis revealed the *Fsp-Cre* transgene targets ~20% of those aSMA+pSTAT3+ fibroblasts, indicating we targeted ~50% of aSMA+pSTAT3+ cells using this strategy (Fig 2D). We performed a similar analysis using PDGFRB as a fibroblast marker and obtained similar results (Fig S3A and B).

Previous work from the Tuveson lab has demonstrated that in the KPC model the intensity of aSMA staining and location relative to tumor cells defines two groups of CAFs: myCAFs are aSMA$^{Hi}$ and tumor proximal, whereas iCAFs are aSMA$^{Lo}$ and distal to tumor (Ohlund et al, 2017; Elyada et al, 2019). We observe the same distribution of aSMA$^{Hi}$ and aSMA$^{Lo}$ CAFs in KPF mice (Fig S3E). Similar analysis of KPF-TF mice revealed that an average of 68% were Tomato$^+$aSMA$^{Lo}$ and 32% were Tomato$^+$aSMA$^{Hi}$, suggesting the transgene targets predominantly iCAFs (Fig 2E, Ohlund et al, 2017; Elyada et al, 2019). Because the endogenous *FSP1/S100A4* gene can be expressed in both fibroblasts and macrophages (Österreicher et al, 2011), we performed an F480/Tomato dual stain analysis on the KPF-TF mouse to demonstrate the specificity of our *Fsp-Cre* transgene in this study (Fig S3C). This analysis revealed no overlap between F480 and Tomato staining (Fig S3D).

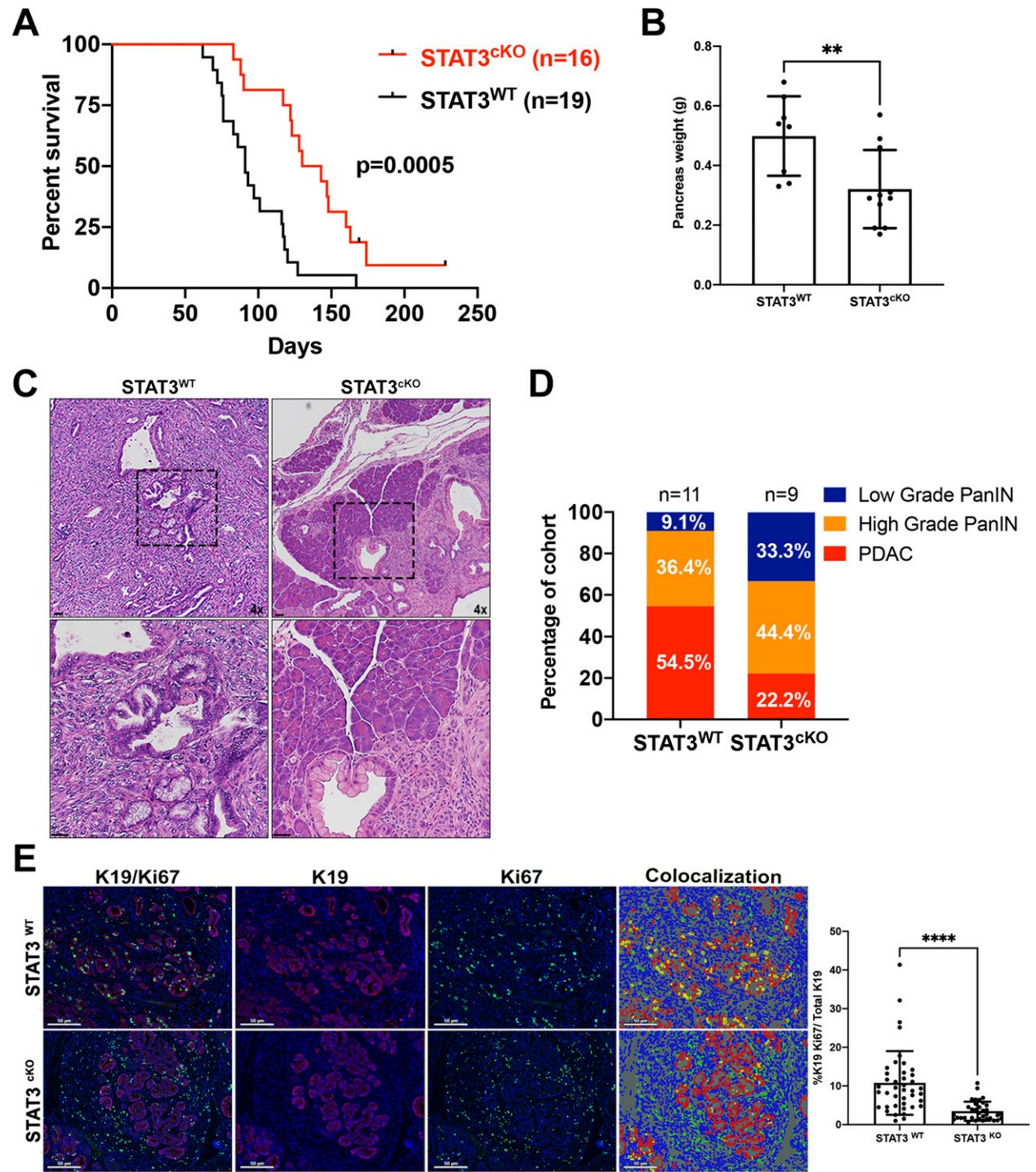

**Figure 3.  Stromal STAT3 drives PDAC tumorigenesis and reduces survival in a Kras^G12D GEMM.**
**(A)** Kaplan–Meier curve depicting survival rate of STAT3^WT (*n* = 19) and STAT3^cKO (*n* = 16) mice. **(B)** Weight of pancreas tumors around 90–100 d of age taken at time of necropsy (STAT3^WT: *n* = 8, STAT3^cKO: *n* = 11). **(C, D)** Representative H&E of STAT3^WT and STAT3^KO PDAC lesions around 90–100 d of age with (D) Pathological scoring of pancreatic lesions (STAT3^WT: *n* = 11, STAT3^KO: *n* = 9). **(E)** Dual color IF of CK19 and Ki67 on PDAC lesions from STAT3^WT (*n* = 7) and STAT3^cKO (*n* = 6) with phenotyping analysis (yellow dots indicate colocalization between CK19 and Ki67) and quantification. Error bars represent mean ± SD. Bars 50 μM (**)*P* < 0.01 (****)*P* < 0.0001.

Survival analysis of a cohort of STAT3^WT and STAT3^cKO mice revealed that loss of fibroblast STAT3 gave a significant survival advantage over the STAT3^WT cohort (Fig 3A). Median survival for

STAT3^WT and STAT3^cKO cohorts was 95 and 130 d, respectively. Pancreas tumor weight was also significantly reduced in the STAT3^cKO cohort at 90 d (Fig 3B). We then scored tumors within the

two genetic cohorts at the 90-d timepoint and found that 54.5% of the mice in the STAT3[WT] group had developed invasive carcinoma compared with 22.2% of mice in the STAT3[cKO] cohort, whereas the reverse was seen regarding low grade PanIn lesions (Fig 3C and D).

Consistent with these results, fibroblast *Stat3* deletion significantly decreased epithelial cell proliferation, as evidenced by positive staining for the ductal cell marker CK19 and the proliferation marker Ki67. Dual IF staining of CK19/Ki67 on pancreatic tumor tissues from the survival cohort revealed that STAT3[cKO] mice had a threefold decrease in Ki67+ ductal cells compared with the STAT3[WT] tumors (Fig 3E).

### CAF STAT3 modulates ECM remodeling

The complex stroma generated by CAFs is composed of collagen structures that influence cancer cell activity and prognosis (Drifka et al, 2016). Consistent with a role for STAT3 in ECM remodeling, Picrosirius Red staining revealed a 32% decrease in overall collagen in the STAT3[cKO] cohort (Fig 4A). Masson's Trichrome staining confirmed a significant decrease in collagen in the STAT3[cKO] cohort compared with the STAT3[WT] (Fig S4A). To address whether this difference in ECM was due to reduced number or reduced activity of CAFs in the STAT3[cKO] cohort, immunofluorescent staining of aSMA was performed and quantified (Fig 4B). This revealed a 36% decrease in total aSMA+ CAFs within the stroma of the STAT3[cKO] cohort compared with the STAT3[WT] (Fig 4C).

To determine whether loss of STAT3 in CAFs affected proliferation or apoptosis in CAFs, we performed dual aSMA: Ki67 (Fig 4B) and dual aSMA: cleaved-caspase 3 (CC3) staining (Fig S4B). In contrast to the tumor cells, no significant change in aSMA+Ki67+ double-positive fibroblasts between the two genetic cohorts was observed (Fig 4C). There was also no significant change in aSMA+CC3+ CAFs between the two cohorts (Fig S4B).

### CAF STAT3 promotes an immunosuppressive TME in PDAC

Most PDAC tumors have an immunosuppressive microenvironment that is not conducive to infiltration and activation of anti-tumor T cells (Karamitopoulou, 2019). Furthermore, many cases of PDAC are associated with increased infiltration of regulatory T cells (Tregs), which can block the anti-tumor activities of effector CD8[+] T cells (Hiraoka et al, 2006; Karamitopoulou, 2019). Our group previously found that loss of the Hedgehog signaling factor Smoothened (SMO) in pancreatic CAFs results in an RNF5-mediated proteasomal degradation of PTEN (Pitarresi et al, 2018). Interestingly, loss of PTEN in CAFs resulted in a significant decrease in CD3[+] T-cell infiltration in the SMO knockout model compared with the WT model (Liu et al, 2016; Pitarresi et al, 2018). Foxp3+ Treg cell populations were also increased in the CAF SMO knockout mice (Liu et al, 2016). We performed CD3/CD8 immunofluorescent staining to assess the presence of CD8[+] cytotoxic T cells around neoplastic lesions between the STAT3[WT] and STAT3[cKO] mice (Fig 5A). We found a significant increase in overall CD3[+] T-cell infiltration in the STAT3[cKO] (average of 6.8% CD3[+] T cells/total cells) compared the STAT3[WT] cohort (3.5% CD3[+] T cells/total cells, Fig 5C). Notably, there was a significant increase in CD8[+] T-cell infiltration in the STAT3[cKO] cohort (1.7% CD3[+]CD8[+] T cells/total cells compared with the STAT3[WT] cohort

[0.89%/total cells Fig 5D]). We next investigated the presence of Tregs using CD4/Foxp3 dual immunofluorescent staining (Fig 5B). Whereas there was no significant difference in overall CD4[+] T-cell infiltration between the two cohorts (Fig 5E), we observed a significant decrease in the ratio of Tregs infiltration in the STAT3[cKO] (14.5% of CD4[+]) cohort compared with the STAT3[WT] (20.24% of CD4[+]) (Fig 5F).

Tumor-associated macrophages are one of the major immune cell populations associated with pancreatic cancer and are important in establishing an immunosuppressive TME early in pancreatic cancer progression (Lankadasari et al, 2019; Liu et al, 2019). Because STAT3 regulates the transcription of many immunomodulatory cytokines, we investigated the presence of phenotypically defined M2 macrophages in the TME within the STAT3[WT] and STAT3[cKO] cohorts. F480 and CD206 dual color IF was used to detect M2 macrophages in the TME between the two genetic cohorts (Fig 6A). The Vectra multispectral platform was used to phenotype and quantify the percentage of F480+CD206+ double-positive M2-like macrophages (see the Materials and Methods section). Overall, we observed no significant change in overall number of F480+ macrophage infiltration between the two cohorts (Fig 6C). Strikingly, comparing F480+CD206+ M2-like macrophages revealed a significant decrease in this population in the STAT3[cKO] genetic cohort compared with STAT3[WT] controls (40% versus 68% F480+CD206+/F480+, respectively, Fig 6D). To determine whether there was a reciprocal change in M1 macrophage populations as a result of CAF STAT3 deletion, F480+ CD86[+] (a marker for M1-like cells, [Orecchioni et al, 2019]) dual color IF was performed on the same samples. This staining demonstrated a switch in macrophage polarization between STAT3[cKO] and STAT3[WT] genetic groups (32% versus 16% F480+CD86+/F480+, respectively, Fig 6B and E). Given the observed change in the ratio of M1:M2 macrophages, we hypothesized that STAT3 signaling from CAFs is promoting the M2-like polarization, but not recruitment, of macrophages within the TME.

### STAT3 in CAFs drives an immunomodulatory secretome

To investigate whether STAT3 signaling in pancreatic CAFs can directly influence macrophage polarization, we performed an in vitro polarization assay on RAW 264.7 macrophages using conditioned medium (CM) collected from WT and STAT3 KO mouse CAFs (confirmation of STAT3 KO in mouse CAFs shown in Fig S5A and B). To measure macrophage polarization, we assessed the gene expression of established M1 (*Nos2*, *Tnf-α*, and *Il-15*) and M2 (*Arg1*, *Mrc1*, and *Il10*) markers (Orecchioni et al, 2019). Initially, a set of control experiments in which we polarized RAW264 cells to either M1-like by LPS and IFN-γ treatment or M2-like using IL-13 and IL-4 treatment, were performed to validate the assay (Fig S5C). Once we established the procedure, we added treatment conditions in which we applied WT CAF CM or KO CAF CM to the cells. We measured a significant increase in the expression of all three M2 markers (Arg-1, Mrc1 [Cd206], and Il10) in the STAT3 WT CM treated macrophages (Fig 7A). However, we found no change in M1-like macrophage marker genes upon treatment with conditioned medium (Figs 7A and S5D).

To investigate which secreted factor(s) from the STAT3 WT CAFs could affect M2-like macrophage polarization, we used a cytokine/

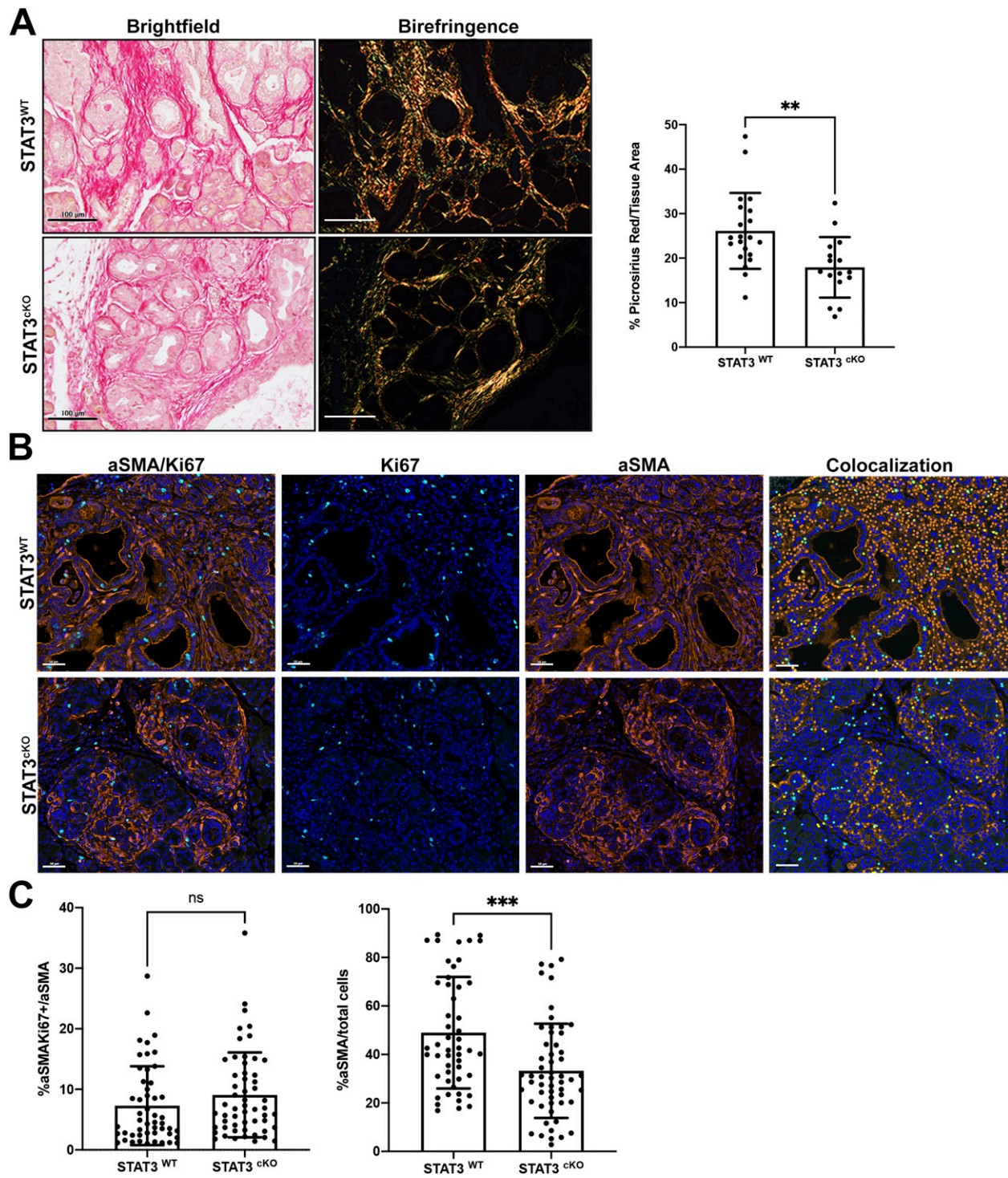

**Figure 4. Cancer-associated fibroblast STAT3 modulates ECM remodeling.**
**(A)** Picrosirius Red staining and quantification on PDAC tumors from STAT3[WT] (*n* = 7) and STAT3[cKO] (*n* = 5) cohorts. **(B, C)** Dual color IF of aSMA (orange) and Ki67(teal) on PDAC tumors from STAT3[WT] (*n* = 4) and STAT3[cKO] (*n* = 5) cohorts with phenotyping analysis (yellow dots indicate cells double-positive for aSMA and Ki67) and (C) quantification. Bars 50 *µ*M. Error bars represent mean ± SD. Bars 100 *µ*M. (**)*P* < 0.01 (***)*P* < 0.001.

chemokine antibody array (R&D Systems) to measure expression of a set of secreted immune-modulatory factors in the conditioned medium from STAT3 KO, and WT CAFs. In addition, because STAT3 is constitutively activated in PTEN-null CAFs (Fig 1), we included those cells expecting that the key factors(s) might be overexpressed compared with WT control.

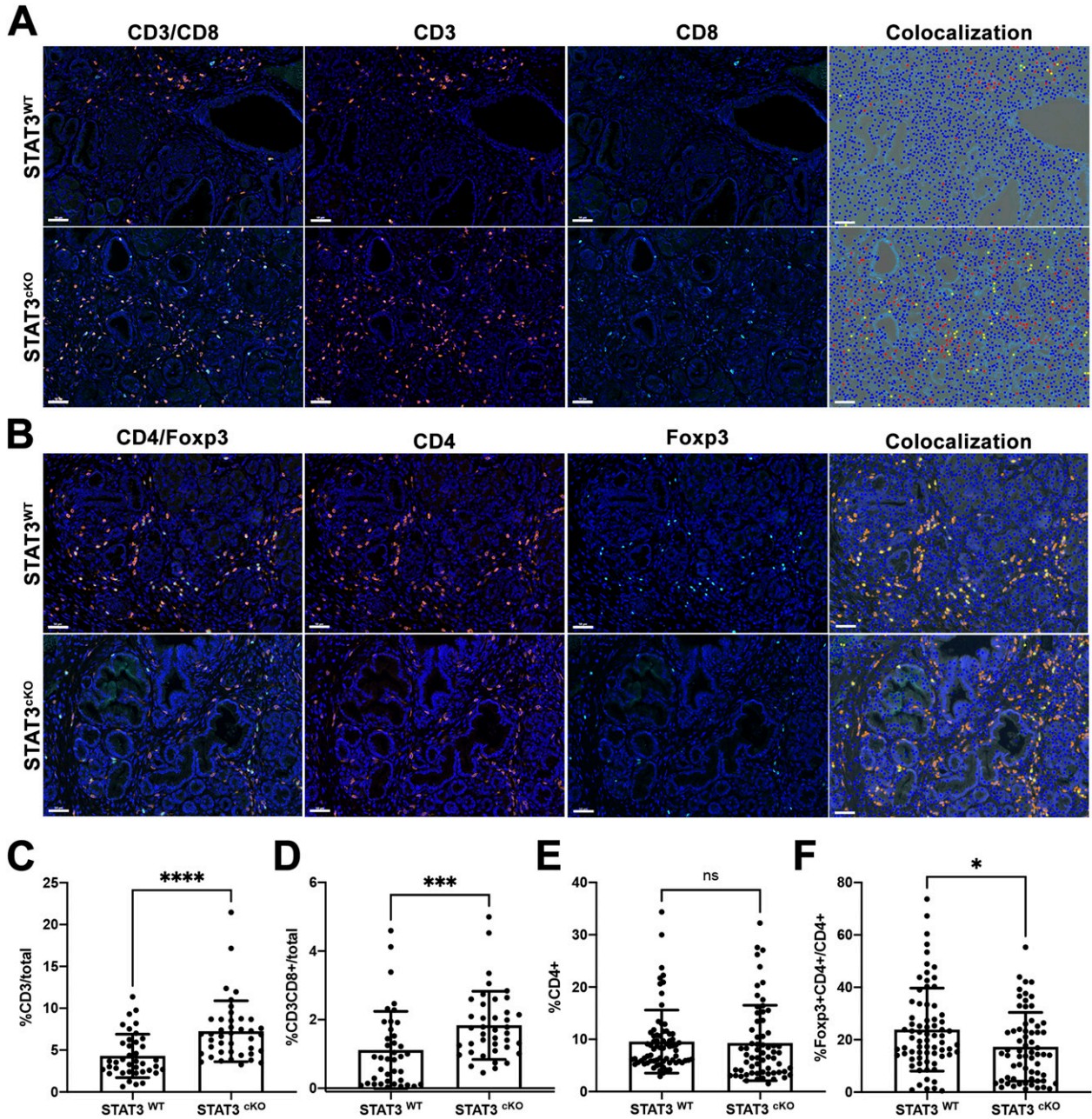

**Figure 5. Loss of fibroblast STAT3 results in an increase in T-cell infiltration in KPF tumors.**
**(A)** Dual color IF of CD3 (orange) and CD8 (teal) on PDAC tumors from STAT3^WT (*n* = 5) and STAT3^cKO (*n* = 6) mice. **(B)** Dual color IF of CD4 (orange) and Foxp3 (teal) on PDAC tumors from STAT3^WT (*n* = 5) and STAT3^cKO (*n* = 6) mice. **(A, B)** Phenotype analysis generated by inFORM software is depicted in the fourth panel; yellow dots indicate double-positive cells for CD3/CD8 (A) or CD4/Foxp3 (B). **(C, D, E, F)** Quantification of IF using inFORM software for phenotyping and quantification of multispectral images. Error bars represent mean ± SD. Bars 50 μM. (*)$P < 0.05$ (***)$P < 0.001$ (****)$P < 0.0001$.

As predicted, we observed a significant decrease in five factors (CCL5, CXCL1, CXCL2, CXCL10, and IL6), in STAT3 KO CM compared with WT CM, and a significant increase in the same cytokines in PTEN KO CM compared with WT CM (Figs 7B and S5E). We performed quantitative real-time RT-PCR on the top ligands to confirm the results of the cytokine array and found that mRNA levels of CXCL1, CXCL10, CCL5, and IL-6 were significantly decreased in PTEN KO fibroblasts compared with WT (Fig S5F) and STAT3 KO fibroblasts compared with WT (Figs 7C and S5G). In addition, an ELISA

performed for CXCL1 confirmed a significant decrease in CXCL1 concentration between PTEN KO versus STAT3 KO CM and STAT3 WT versus STAT3 KO CM (Figs 7D and S5H).

### CAF STAT3 promotes M2 polarization through the secretion of CXCL1

Given CXCL1 was one of the most down-regulated cytokines in the STAT3 KO CAF conditioned medium and the most up-regulated factor in the PTEN KO CAF CM, we hypothesized this cytokine could

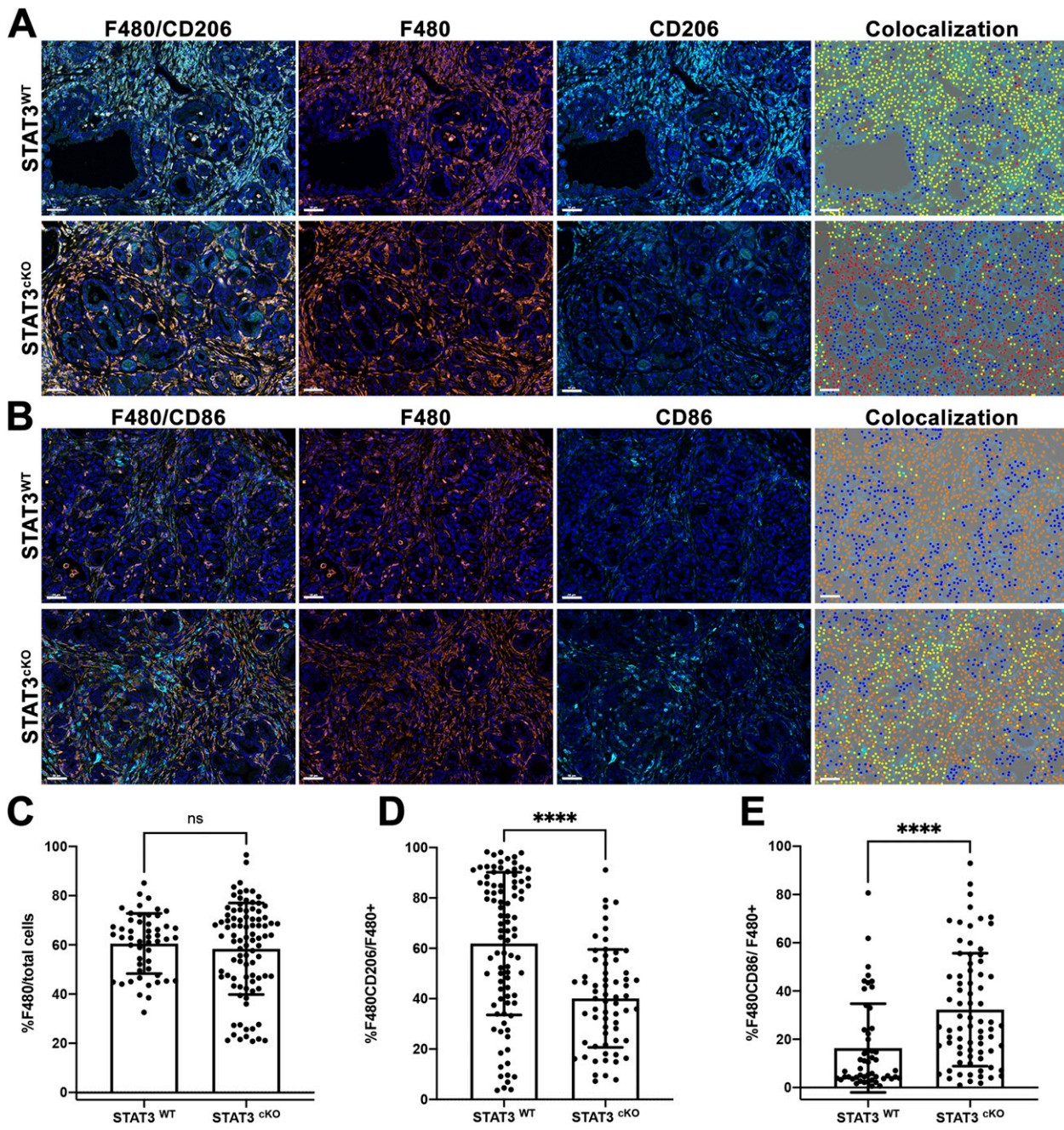

**Figure 6. Depletion of cancer-associated fibroblast STAT3 results in a decrease in M2 macrophages and an increase in M1 macrophages in KPF tumors.**
**(A)** Dual color IF of F480 (orange) and CD206 (teal) on PDAC tumors from STAT3^WT ($n$ = 5) and STAT3^cKO ($n$ = 6) mice. **(B)** Dual-color IF of F480 (orange) and CD86 (teal) on PDAC tumors from STAT3^WT ($n$ = 5) and STAT3^cKO ($n$ = 6) mice. **(A, B)** Phenotype analysis generated by inFORM software is depicted in fourth panel; yellow dots indicate double-positive cells for F480/CD206 (A) or F480/CD86 (B). **(C, D, E)** Quantification of IF using inFORM software for phenotyping and quantification of multispectral images. Error bars represent mean ± SD. Bars 50 $\mu$M (****)$P$ < 0.0001.

be responsible for influencing macrophage polarization in our in vivo model. To test whether CXCL1 secreted from pancreatic WT CAF conditioned medium is responsible for inducing M2 polarization, we repeated the macrophage polarization assay but with the addition of a CXCL1 neutralizing antibody to the WT conditioned medium. Addition of the antibody significantly decreased the gene expression of M2 markers *Arg1*, *Mrc1* and *Il10* compared with WT

condition medium treatment alone (Fig 7E). Interestingly, addition of the antibody significantly increased the expression of M1 marker *Nos2* (Fig 7E). We also investigated the effect of inhibiting CXCR2, the receptor for CXCL1. Using a CXCR2 inhibitor (SB225002), we found the expression of M2 genes *Il10*, *Mrc1*, and *Arg1* to be significantly decreased, whereas the M1 marker *Nos2* was significantly increased (Fig 7F). We then performed a rescue experiment in which we added

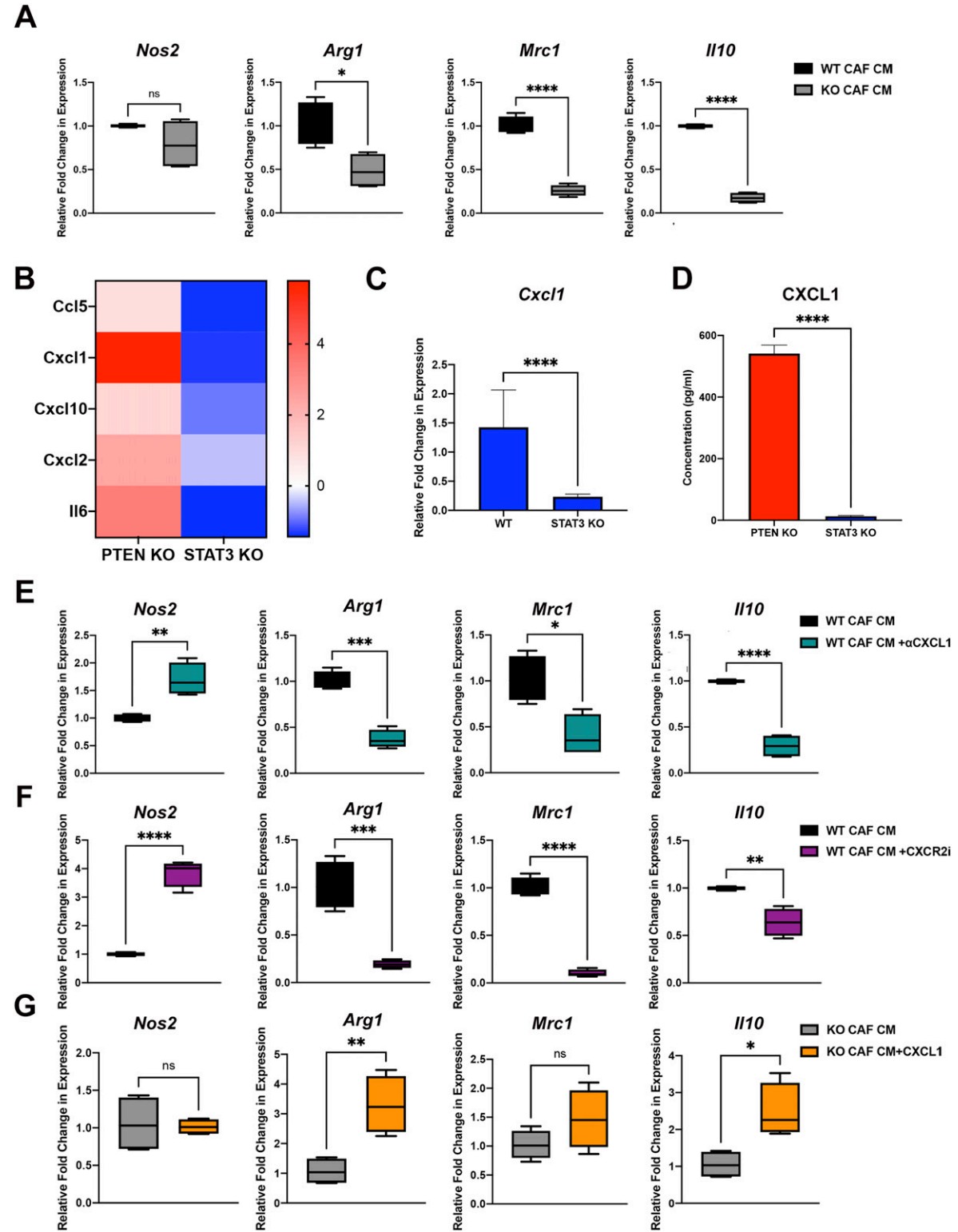

**Figure 7. STAT3 signaling in cancer-associated fibroblasts (CAFs) promote an immunomodulatory secretome.**
**(A)** Relative gene expression for M1 and M2 macrophage genes in RAW 264.7 cells treated with WT CAF CM or KO CAF CM. **(B)** Heat map depicting relative protein expression of cytokines in PTEN KO and STAT3 KO CM. **(C)** Relative gene expression of *Cxcl1* in STAT3 WT and STAT3 KO CAFs. **(D)** Concentration of CXCL1 in CM of PTEN KO and STAT3 KO CAF CM. **(E, F, G)** Relative gene expression for M1 and M2 macrophage genes in RAW 264.7 cells treated with WT CM or with WT CAF CM+anti-CXCL1 or (F) CXCR2 inhibitor (10 µM) or (G) KO CAF CM with or without soluble CXCL1 (150 ng/ml). (*)P < 0.05 (**)P < 0.01 (***)P < 0.001 (****)P < 0.0001.

soluble CXCL1 to the STAT3 KO CM. We found there was no change in the expression of *Nos2*; however, the addition of CXCL1 significantly increased the expression of *Arg1* and *Il10* (Fig 7G). Taken together, these results indicate that CXCL1 can directly influence M2 polarization in vitro.

## Discussion

Previous work from our group found that loss of PTEN in pancreatic CAFs resulted in a shift in immune populations within the stroma (Liu et al, 2016). Herein we identify STAT3 as a critical downstream effector of PTEN, which alters the PDAC immune landscape. We demonstrate that genetic ablation of STAT3 in PDAC CAFs prolonged survival and reduced tumor progression in the KPF mouse model. Mechanistically, STAT3 signaling in CAFs facilitated pancreatic tumorigenesis by increasing fibrosis and establishing an immunosuppressive TME.

Our study quite strikingly demonstrates STAT3 signaling from CAFs promotes M2 macrophage polarization within the PDAC TME but does not affect recruitment of F480+ macrophages. Strategies aimed at eliminating or inhibiting tumor macrophage recruitment in PDAC have been explored, but need to be approached with caution, given some macrophage populations have tumor-limiting properties (Ino et al, 2013; Allavena et al, 2021). Thus, there are various strategies proposed to re-polarize macrophages already present in the tumor stroma, and transforming M2 macrophages to the M1 phenotype may be sufficient to limit tumor growth (Allavena et al, 2021). In addition, loss of CAF STAT3 led to an increase in CD8[+] T cells and decrease in Foxp3+ Tregs in tumor lesions, which may be a consequence of the decreased number of M2 macrophages, or alternatively indicate direct communication between CAFs and T-cell subsets. Although priming of the T-cell response relies heavily upon macrophages, some studies have shown CAFs could directly affect T-cell activity through expression of immune checkpoints such as PD-L1 and the secretion of soluble factors such as COX-2 and TGF-B (Gorchs & Kaipe, 2021). Our results suggest targeting CAFs may provide an alternative strategy to "re-educate" macrophages to reverse immunosuppression and increase T-cell recruitment into tumors. STAT3 signaling in fibroblasts may promote the polarization of M2 macrophages through increased secretion of the cytokine CXCL1. To our knowledge, CXCL1 has not previously been described as a direct target of STAT3. CXCL1 has already been identified as one secreted factor from CAFs that can promote tumor progression (Biffi et al, 2019; Manoukian et al, 2021). Importantly, CXCL1 is up-regulated in human PDAC compared with normal pancreas (Matsuo et al, 2009). In addition, several studies have demonstrated that CXCL1 signaling can promote M2 polarization both in vitro and in vivo (Li et al, 2018; Sano et al, 2019; Le Naour et al, 2020). Our studies suggest CXCR2 inhibitors may provide a means to increase the presence of M1 macrophages in PDAC in vivo, a hypothesis to be tested in future experiments.

The results here provide genetic evidence that these STAT3+ CAFs comprise a subpopulation crucial to the establishment of an environment permissive to tumor development and progression. Strikingly, our analysis of the KPF-TF model indicated that the FSP-Cre transgene used targeted only ~20% of overall aSMA+ CAFs in the

STAT3[cKO] model, which was still sufficient to reveal the dramatic differences in survival and tumor characteristics compared with KPF alone. Consistent with these results, STAT3 was targeted predominantly in the tumor-promoting iCAF subpopulation defined by the Tuveson lab (Ohlund et al, 2017; Elyada et al, 2019), Interestingly, a portion of the tumor-proximal myCAFs were also targeted. Consistent with this result, we observed a decrease in collagen and extracellular matrix production when STAT3 was deleted in CAFs. Because myCAFs express higher levels of collagens and extracellular matrix components (Elyada et al, 2019), this suggests that STAT3 could play distinct roles in myCAFS compared with iCAFs. Recent work has accentuated that CAF heterogeneity is a key variable in determining tumor aggressiveness (Ozdemir et al, 2014; Kalluri, 2016; Ohlund et al, 2017). The PTEN–STAT3 axis described herein defines a mechanism by which CAFs become protumorigenic in the PDAC TME. The significant increase in overall survival in the STAT3[cKO] model demonstrates the profound influence STAT3+ stromal fibroblasts have on mediating tumor progression and the immune microenvironment.

To our knowledge, the PTEN-STAT3 axis has not been described in CAFs. However, the PTEN/PI3K pathway has previously been found to regulate STAT3 activation in other contexts. For example, constitutive PI3K activation leads to STAT3-Y705 phosphorylation through TEC family non-receptor tyrosine kinases (Hart et al, 2011; Vogt & Hart, 2011), and PTEN loss is correlated with TEC-dependent STAT3 activation in glioblastoma (Guryanova et al, 2011; Vogt & Hart, 2011; Moon et al, 2013). Our results suggest that the PI3K pathway may not be driving constitutive STAT3 phosphorylation, and instead, PTEN loss may lead to direct activation of RTK signaling or up-regulation of growth factors and cytokines. In turn, these factors may act in an autocrine fashion to stimulate STAT3 phosphorylation.

Our results identify an unexplored role for CAF STAT3 in altering ECM production and composition. Previous studies from our group demonstrated loss of PTEN in the breast cancer stroma results in an increase in ECM deposition and remodeling (Trimboli et al, 2009; Jones et al, 2019), suggesting STAT3 could be involved in modulating the ECM (Ray et al, 2013). Targeting STAT3 to remodel ECM could hold therapeutic promise. Nagathihalli et al (2015) previously described the JAK inhibitor AZD1480 combined with gemcitabine-improved drug delivery in GEM models of PDAC because of enhanced remodeling of the tumor stroma (Nagathihalli et al, 2015). More recently, Src/EGFR inhibition was found to target STAT3 and remodel the tumor stroma, enhancing delivery of gemcitabine in a GEM model of PDAC (Dosch et al, 2020). Future studies aimed at identifying how STAT3 in CAFs promotes collagen production or organization are warranted.

Whereas targeting STAT3 remains a clinically promising target for solid tumors including PDAC, only a few clinical trials have shown favorable outcomes (Peisl et al, 2021). However, many of these clinical studies do not examine the influence of the various STAT3 inhibitors on the stroma or immune cells (Peisl et al, 2021). Despite many advances in the field of cancer immunotherapy, PDAC remains resistant to approved immunotherapeutic strategies, such as checkpoint inhibition therapy (Royal et al, 2010; Brahmer et al, 2012). Our reported increase in CD8[+] T-cell infiltration in PDAC tumors in the STAT3[cKO] cohort indicate checkpoint therapy could be more

efficacious as a result of loss of stromal STAT3. Some preclinical studies demonstrating the promise of combining STAT3 inhibitors and immune-checkpoint therapy have led to multiple clinical trials incorporating this strategy in advanced solid tumors (Lu et al, 2017; Zou et al, 2020). Our study indicates that STAT3 inhibition could work in two distinct ways: (1) decrease the fibrotic stroma and (2) limit immunosuppressive pathways.

In conclusion, our study revealed a tumor-promoting mechanism in CAFs activated by disruption of PTEN signaling and identified potential targets for restoring the anti-tumor immune response in PDAC.

# Materials and Methods

### Animal strains and maintenance

All animal studies were conducted in compliance with federal and university standards. Use of animals was approved by the Ohio State University and Medical University of South Carolina IACUCs (Protocols: 2007A0120-R3 and IACUC-2017-00065; PI: MCO). *FSP-Cre* mice were previously described (Trimboli et al, 2008, 2009). *Pdx1-Flp* knock-in mice were also previously described (Wu et al, 2017). *FSF-Kras$^{G12D/+}$* mice were provided by Tyler Jacks. *Trp53$^{frt/+}$* and *Rosa$^{LSL-FLF-GFP}$* mice have been described previously (Wu et al, 2017). *Stat3$^{LoxP}$* mice were obtained from Jackson Laboratories. Animals were on a mixed C57Bl/6; 129Sv; Black Swiss background. *Pdx1FlpO$^{ki}$*, *FSF-Kras$^{G12D/+}$*, *Trp53$^{frt/+}$*, and *Stat3$^{fl/fl}$* were all interbred to obtain control and experimental PDAC mice (mice were monitored weekly) and both control (*PdxFlpO$^{ki/+}$; FSF-Kras$^{G12D/+}$; Trp53$^{frt/frt}$; Stat3$^{fl/fl}$*) and experimental (*PdxFlpO$^{ki/+}$; FSF-Kras$^{G12D/+}$; Trp53$^{frt/frt}$; FSP-Cre; Stat3$^{fl/fl}$*) mice were removed at specified time points or when they met Early Removal Criteria. The overall health of the animals was monitored by trained laboratory personnel and the veterinary staff.

### Fibroblast isolation and cell culture

Primary murine and human pancreatic fibroblasts were isolated as previously described (Liu et al, 2016). PTEN$^{cKO}$ cultures were established from isolated pancreatic fibroblasts that carried *FSP-Cre;Pten$^{LoxP}$* alleles as described previously (Pitarresi et al, 2018). *Pten* knockdown in human PDAC fibroblasts was achieved using Dharmacon GIPZ lentiviral shPTEN (Horizon Discovery, Ltd.) following the manufacturer's instructions. Briefly, fibroblasts isolated from two independent human PDAC specimens were seeded in a six-well dish. To each well, 1 ml serum-free DMEM containing 1× polybrene and 0.3 μl lentivirus stock was added. After 6 h, 1 ml complete medium was added. GFP+ fibroblasts were isolated and transferred to t25 cell culture flasks once confluent. STAT3$^{WT}$ and STAT3$^{cKO}$ cultures were established from the pancreatic tumor of 4-mo-old male KPF Stat3$^{fl/fl}$ mice. Deletion of *Stat3* was achieved ex vivo using a CRE (GFP-Puro), CAG lentivirus (LVP576; GenTarget). Briefly, cells were plated at 70% confluency in a 24-well plate in DMEM supplemented with 10% FBS. 50 μl of the virus stock (1 × 10$^7$ IFU/ml in DMEM with 10% FBS and 10× polybrene) was added directly to fresh 0.5 ml of complete medium and that mixture was

added to the cells. Antibiotic selection started 72 h after initial transduction (1 μg/μl). Fibroblasts went through two rounds of clonal selection to isolate STAT3 KO colonies. Three separate clones were used as biological replicates in subsequent in vitro experiments. Unless otherwise stated, cells were maintained in high-glucose DMEM supplemented with 10% FBS and 1% penicillin–streptomycin in a humidified incubator in 5% CO$_2$ at 37°C.

### Inhibitor studies

Primary murine WT and PTEN KO CAFs were serum starved overnight before treatment. The next day, inhibitors were added in indicated concentrations (Figs 1 and S1) in serum-free medium and treated for 4 h. Cells were then stimulated with 10% before lysing and protein was collected for downstream Western blotting. The following inhibitors were used: Momelotinib (JAK inhibitor) (Cat. no. S2219; Selleckchem), Pictilisib (pan PI3K inhibitor) (#S1065; Selleckchem), Alpelisib (PI3K-α) (#S2814; Selleckchem), and IPI-3406 (PI3K-γ) (#S7335; Selleckchem). Each experiment was repeated three times.

### IHC and IF staining

Mouse pancreas tissue was harvested at the time of euthanasia and fixed in 10% neutral-buffered formalin solution. After formalin fixation for 24 h, pancreatic tissue was placed in 70% ethanol, embedded in paraffin, and cut into 4-μm sections. Tissues were deparaffinized, rehydrated, and stained with H&E, Masson's Trichrome, or Picrosirius Red (Polysciences, Inc.). H&E slides were assessed for tumor grade/differentiation. For autostaining, unstained slides with 4- to 5-μm sections of FFPE tissue were baked at 65°C for 15 min and automated software performed dewaxing, rehydration, antigen retrieval, blocking, primary antibody incubation, post primary antibody incubation, detection and counterstaining. Samples were then removed from the autostainer and dehydrated through use of ethanol and xylene, then mounted and coverslipped.

Optimized chromogenic multiplex was performed using the Roche multiplexing method. Roche multiplex protocol involves using IHC multiplexing protocols for various markers to show presence and spatial relationship using the Roche Ventana Discovery Ultra Automated Research Stainer (Roche Diagnostics). Tissues were stained with antibodies against pSTAT3 (Tyr705) (1:200; Cat. no. 9145; Cell Signaling), PTEN (1:100; Cat. no. 138G6; Cell Signaling), aSMA (1:100Cat. no. 19245S; Cell Signaling), and the signals were generated using the following chromogens: purple (760229) and yellow (760-239) (Roche Diagnostics).

Optimized multiplex immunofluorescence was performed using the OPAL multiplexing method. OPAL is based on tyramide signal amplification (TSA) using the Roche Ventana Discovery Ultra Automated Research Stainer (Roche Diagnostics). The following antibodies were used to detect specific proteins within each section: anti-aSMA (1:2,000; Cat. no. 19245S; Cell Signaling), anti-pSTAT3 (Tyr705) (1:200; Cat. no. 9145; Cell Signaling), anti-CD3 (1:100; Cat. no. 78588S; Cell Signaling), anti-Ki67 (1:400; Cat. no. ab16667; Abcam), anti-CD8a (1:50; Cat. no. 98941; Cell Signaling), anti-CD86 (1:100; Cat. no. 19589S; Cell Signaling), anti-F4/80 (1:100; Cat. no. 70076S; Cell

Signaling), anti-CD4 (1:100; Cat. no. ab183685; Abcam), anti-CD206 (1:200; Cat. no. NBP1-90020; Novusbio), anti-K19 (1:50; Cat. no. DSHB; TROMA III), anti-Stat3 (1:300; Cat. no. 9139S; Cell Signaling), anti-Tomato (1:1,000; Cat. no. 600-401-379S; Rockland), anti-Foxp3 (1:50; Cat. no. 12653; Cell Signaling), anti-Pdgfrb (1:100; Cat. no. 3169S; Cell Signaling), and anti-PTEN (1:200; Cat. no. 138G6; Cell Signaling). The fluorescence signals were generated using the following fluorophores: OPAL 480, 620, 690, 520, and 570 (Akoya Biosciences). DAPI was used for nuclear counterstaining. Between each sequential antibody staining step, slides were incubated in citrate buffer pH 6 (Cell Conditioning Solution [CC2] Cat. no. 980-223; Roche Diagnostics) at 90 °C for 8 min to remove the previous primary and secondary antibody complexes. Stained slides are mounted with ProLong Gold Antifade Reagent (Cat. no. P36934; Thermo Fisher Scientific) and imaged using the Akoya Vectra Polaris Automated Imaging system (Akoya Biosciences).

## Pancreas histopathology

H&E-stained sections of tumor samples were assessed in consultation with two veterinary pathologists using the recommended nomenclature for pancreas exocrine neoplasia (Hruban et al, 2006). Because of the heterogeneous nature of each tumor, each sample per cohort was graded based on the majority lesion present. For example, Low Grade PanIN designation was defined as majority PanIN 1-1B, High Grade PanIN designation was defined as majority PanIN 2–3, and PDAC was designated if there was an overwhelming presence of invasive carcinoma.

## Multispectral vectra IHC analysis

Immunostained samples were imaged using Vectra Polaris Automated Quantitative Pathology Imaging System (PerkinElmer, Inc.). The acquisition workflow has been described elsewhere (Pitarresi et al, 2016). For quantification of immunofluorescent staining, at least three multispectral images per animal with at least three mice per genotype were acquired at 20× magnification. To control for the heterogeneous nature of pancreas tumors from both STAT3[WT] and STAT3[cKO] cohorts, stamped regions of similar histopathology in 90-d-old mice were used for immunofluorescent quantification in Figs 4–6. PerkinElmer's inForm Software v2.4.10 (Akoya Biosciences) software was used to spectrally unmix images and analyze the staining. Algorithms were made to segment images into individual cells using DAPI nuclear signal and a defined outer distance to infer cytoplasm and membrane. Machine learning was used to phenotype each cell into different cell types based on marker gene expression. Relative cell numbers were calculated by normalizing each category of cells using the sum of all categories. The same algorithm was applied to all images across the two genetic cohorts.

Bright-field images of Picrosirius Red (PSR) staining were acquired on a Nikon Eclipse microscope equipped with a Nikon DS-Ri2 camera. A polarized lens attachment was used for analyzing birefringence. Images were captured with the same parameters (i.e., the same light intensity and angle of the polarizing lens 90°C to the light source). ImageJ (FIJI) software was used to quantify both bright-field and polarized images. Briefly, color spectra were separated either by color deconvolution (for brightfield images) or

red/green (for polarized images) and thresholds were set. Data are given as percent of the thresholded area. Bright-field images of Masson's Trichrome staining were acquired as described above. ImageJ (FIJI) software was used to quantify the blue area as previously described (Sarma et al, 2010). Data are given as a percent of the thresholded area.

## Macrophage polarization assays

CAFs were cultured in 10% complete DMEM. When CAFs reached 80% confluency, the medium was removed and cells were washed with PBS. Next, 0.1% FBS DMEM was added to the cells and was left overnight or for about 16 h. Conditioned medium was removed, spun down and aliquoted. RAW 264.7 cells were plated at $2 \times 10^5$ per well in a 24-well tissue culture treated plate and left for 48 h. For M1 polarization conditions, IFN-γ (2 ng/ml) and LPS (100 ng/ml) were added to 10% complete DMEM and added to RAW cells and were treated for 24 h. M2 polarization conditions were the same except RAW cells were treated with IL-4 (50 ng/ml) and IL-13 (50 ng/ml). For the WT and KO CM treatments, RAW cells were treated for 24 h with 50% CM from either WT or KO CM collected from CAFs. Anti-CXCL1 neutralizing antibody (R&D Systems) was added at 150 ng/ml and CXCR2 inhibitor (SB225002; Selleckchem) was added at 15 µM. Soluble CXCL1 (Peprotech) was added at 100 ng/ml. After 24 h of treatment, TRIzol was added for immediate RNA isolation and downstream analyses.

## Cytokine analysis

Conditioned medium from PTEN KO and STAT3 KO pancreatic fibroblasts were analyzed for changes in secreted cytokines using the Proteome Profiler Mouse Cytokine Array Kit (ARY006; R&D Systems, Inc.). Cells were plated in six-well plates with complete growth medium and allowed to attach overnight. The medium was replaced with DMEM supplemented with 0.1% FBS for 16 h before collection. The array was performed according to the manufacturer's instructions. Densitometry/pixel density was captured using the ChemiDoc imaging system (Bio-Rad) and analyzed using ImageJ (Janes, 2015). The mouse CXCL1/KC and CCL5/RANTES Quantikine ELISA (R&D Systems) kits were used to measure secreted CXCL1 and CCL5 in PTEN[cKO] and STAT3[cKO] cell supernatants according to the manufacturer's instructions.

## Western blotting

Cells were lysed in 1x RIPA buffer (Cell Signaling Technology) supplemented with 1× Halt protease and phosphatase inhibitor cocktail (Thermo Fisher Scientific) in accordance with the manufacturer's instructions. Protein concentration was determined via BCA protein assay (Pierce). Equivalent protein from each sample was separated by SDS–PAGE and transferred to polyvinylidene difluoride membrane by wet transfer at 4°C. Blots were washed in 1× TBS with 0.05% Tween-20 (TBST) and blocked for 1 h with 5% milk in TBST at room temperature. The blots were incubated overnight at 4°C with primary antibody (1:1,000) in 5% BSA in TBST, washed, and incubated with secondary antibody (1:5,000) with 5% milk in TBST for 1 h, washed, and developed with ECL Clarity Max (Bio-Rad Laboratories). Primary antibodies used in this study include:

β-Actin (13E5) Rabbit mAb #4970, PTEN (138G6) Rabbit mAb #9559, Stat3 (79D7) Rabbit mAb #4904, Phospho-Stat3 (Tyr705) (D3A7) XP Rabbit mAb #9145, ERK1/2 Rabbit mAb #9102S, pERK1/2 Rabbit mAb #4370S, AKT Rabbit mAb #9272S, pAKT Rabbit mAb #9271S, vinculin Rabbit mAb #05386 (EMD Millipore). Secondary antibody is Peroxidase AffiniPure Goat Anti-Rabbit IgG (H&L) (111-035-003; Jackson Immunoresearch). Densitometries were determined using ImageJ.

### RNA and real-time PCR

RNA was isolated from cells using TRIzol (Invitrogen) or PureLink RNA Mini Kit (Invitrogen) following the manufacturer's instructions. Total RNA (1 μg) was converted to cDNA using the Quantabio qScript cDNA SuperMix. Real-time quantification of mRNA was performed using Taqman (Thermo Fisher Scientific) probes. Applied Biosystems real-time PCR machines were used. Specific probes are listed in the table below:

**TaqMan primers used for real time PCR**

| GapdH | Mm99999915_g1 |
|---|---|
| Cxcl1 | Mm04207460_m1 |
| Tgfb1 | Mm01178820_m1 |
| Il6 | Mm00446190_m1 |
| CCl2 | Mm00441242_m1 |
| Hif1a | Mm00468869_m1 |
| Vegfa | Mm00437306_m1 |
| Il10 | Mm01288386_m1 |
| Tnfalpha | Mm00443258_m1 |
| Arg1 | Mm00475988_m1 |
| Nos2 | Mm00440502_m1 |
| MMP9 | Mm00442991_m1 |
| Mrc1 | Mm01329362_m1 |

### Statistical analysis

Statistical analyses were conducted with Prism 9 (GraphPad Software). $P$-values < 0.05 were considered statistically significant. For all data, normality was assessed by D'Agostino–Pearson omnibus, Shapiro–Wilk, and Kolmogorov–Smirnov normality testing. Data were considered normally distributed upon passing any of the three tests. For pairwise comparisons of normally distributed data, a $t$ test was used. All analyses using $t$ test were two-tailed. A Mann-Whitney U test was used to determine statistical significance for data not normally distributed. A Kaplan–Meir curve was used for the survival study. A one-way ANOVA was used to determine statistical differences between multiple groups.

## Supplementary Information

## Acknowledgements

This study was supported by National Institutes of Health grants P01 CA203653 (DC Guttridge, MC Ostrowski, TA Zimmers), R01 CA208253 (GB Lesinski, MC Ostrowski), T32CA193201 (CB MarElia-Bennett, L Han), and F32CA254328 (L Han) and by American Cancer Society Postdoctoral Fellowship PF-20-114-01-DDC (L Han). This work was also supported by the Hollings Cancer Center Abney Fellowship (L Han and JE Lefler) and by MUSC Hollings Cancer Center Translational Science Shared Resource, Flow Cytometry and Cell Sorting Shared Resource, Cell and Molecular Imaging Shared Resource and the Biostatistics Shared Resource (P30 CA138313). Any opinions, findings, and conclusions expressed in this material are those of the author (s) and do not necessarily reflect those of the HCC Fellowship Program.

### Author Contributions

JE Lefler: conceptualization, data curation, formal analysis, investigation, and writing—original draft, review, and editing.
CB MarElia-Bennett: data curation, formal analysis, investigation, and writing—original draft, review, and editing.
KA Thies: writing—review and editing.
BE Hildreth: writing—review and editing.
SM Sharma: methodology and writing—review and editing.
JR Pitarresi: writing—review and editing.
L Han: writing—review and editing.
C Everett: data curation and writing—review and editing.
C Koivisto: writing—review and editing.
MC Cuitino: methodology and writing—review and editing.
CD Timmers: methodology and writing—review and editing.
E O'Quinn: writing—review and editing.
M Parrish: methodology and writing—review and editing.
MJ Romeo: writing—review and editing.
AJ Linke: writing—review and editing.
GA Hobbs: writing—review and editing.
G Leone: writing—review and editing.
DC Guttridge: writing—review and editing.
TA Zimmers: writing—review and editing.
GB Lesinski: writing—review and editing.
MC Ostrowski: conceptualization, data curation, and writing—original draft, review, and editing.

### Conflict of Interest Statement

GB Lesinski has Grant/Research Support through sponsored research agreements between Emory University and from Merck and Co., Bristol-Myers Squibb, Boerhinger-Ingelheim, and Vaccinex and a consulting or advisory role for ProDa Biotech, LLC. The other authors have declared that no conflict of interest exists.

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
