## [Reviewer comments · Life Science Alliance]

STAT3 in Tumor Fibroblasts Promotes an Immunosuppressive Microenvironment in Pancreatic Cancer

Julia E. Lefler, Catherine B. MarElia-Bennett, Katie A. Thies, Blake E. Hildreth III, Sudarshana M. Sharma, Jason R. Pitarresi, Lu Han, Caroline Everett, Christopher Koivisto, Maria C. Cuitino, Cynthia D. Timmers, Elizabeth O'Quinn, Melodie Parrish, Martin J. Romeo, Amanda J. Linke, G. Aaron Hobbs, Gustavo Leone, Denis C. Guttridge, Teresa A. Zimmers, Gregory B. Lesinski, Michael C. Ostrowski
DOI: 10.26508/lsa.202201460

Corresponding author(s): Dr. Michael C Ostrowski (Medical University of South Carolina)

Review Timeline:

Submission Date:	2022-03-21
Editorial Decision:	2022-04-11
Revision Received:	2022-05-24
Editorial Decision:	2022-06-17
Revision Received:	2022-06-23
Editorial Decision:	2022-06-24
Revision Received:	2022-06-30
Accepted:	2022-06-30

Transaction Report:

April 11, 2022

Re: Life Science Alliance manuscript #LSA-2022-01460-T

Dr. Michael C Ostrowski
Medical University of South Carolina
Hollings Cancer Center
86 Jonathon Lucas Street
HO 512D
Charleston, SC SC 29425

Dear Dr. Ostrowski,

Thank you for submitting your manuscript entitled "STAT3 Signaling in Cancer-Associated Fibroblasts Promotes an Immunosuppressive Microenvironment in Pancreatic Cancer" to Life Science Alliance. The manuscript was assessed by expert reviewers, whose comments are appended to this letter. We invite you to submit a revised manuscript addressing the Reviewer comments.

Thank you for this interesting contribution to Life Science Alliance. We are looking forward to receiving your revised manuscript.

Sincerely,

B. MANUSCRIPT ORGANIZATION AND FORMATTING:

Reviewer #1 (Comments to the Authors (Required)):

In the present work authors investigate pancreatic ductal adenocarcinoma (PDAC) tumor promoting pathways in cancer-associated fibroblasts (CAFs) and identify PTEN-STAT3 axis in pancreatic fibroblasts as a novel mechanism in the modulation of the immune-suppressive tumor microenvironment (TME). This study addresses a relevant topic in the field with important implications for the development of new therapeutic strategies to target stromal interactions in solid tumors. Here, authors demonstrate the involvement of STAT3 signaling in the polarization of macrophages and infiltration of CD8+ T cells affecting the PDAC immune landscape and limiting tumor progression by knocking-down STAT3 in fibroblasts in a KRAS-induced PDAC mouse model.

The authors present robust and consistent data showing that loss of the tumor suppressor PTEN in pancreatic CAFs lead to an enhancement of the STAT3 signaling and secretion of CXCL1 which favoured M2 polarization providing a potential mechanism by which CAFs promote an immune-suppressive TME. In addition, authors demonstrate that depletion of STAT3 in a combined KPF mouse model with fibroblast-specific Cre mediated conditional STAT3 KO slows down tumor progression and increases survival setting the rationale for the development of new therapies for the treatment of pancreatic cancer. In this view, the identification of STAT3 as a downstream effector of PTEN opens new avenues for CAF and immune modulation using CXCR2 and/or JAK inhibitors.

The study is well designed where controls, number of replicates, material (antibodies, etc), and statistical analysis are specified. Manuscript is well written and accurate (ref, etc) and conclusions address limitations and future directions demonstrating a high level of expertise in the field with proven references to previous publications in the area that are consistent with the results and conclusions of the present work. The results are robust and have been confirmed in multiple independent experiments and using different techniques highlighting the high quality of the study. Interpretation of the results are clear and concise and show no inconsistencies.

General comments:

1.- Findings pointed to an upregulation of STAT3 in SMA+ CAF cells; nonetheless, the role of IL6-STAT3 axis in the modulation of different CAF populations (i.e. myCAF and iCAF) still needs clarification. As stated by the authors, the identification of CAFs subtypes within the PDAC TME with both tumor-promoting and tumor-limiting characteristics emphasizes the need of a better understanding of the CAF contribution to the PDAC pathogenesis.

Therefore, I would point out that the problem of CAF subtype identity could be better addressed. The authors are aware of the myCAF and iCAF subtypes, for example, however their use of aSMA as a CAF marker more specifically examines myCAFs rather than iCAFs. In this study, then, associations between markers and aSMA expression are examining their association with myCAFs rather than any other. This does not invalidate the findings of the study, and indeed the correlation of those findings with PDGFRb is good (since there are PDGFRb-expressing iCAFs and myCAFs). It would be good to see addressed this limitation of using aSMA and how this may alter the findings in the study. For example, the effects of STAT3 ablation on collagen deposition may be more relevant to myCAFs, whereas iCAFs are associated with secretion of inflammatory cytokines. Does STAT3 have different roles in different CAF subtypes?

2.- To further confirm the relationship between PTEN and activated STAT3, authors have assessed whether P-STAT3 was increased in PTEN KO fibroblasts. PTEN is a known negative regulator of STAT3 which depletion in CAFs induced increased P-STAT3 as expected (fig1). Further mechanistic studies are needed to elucidate whether the effect comes from the canonical RTK/JAK or non-canonical PI3K/TEC pathways as stated by the authors. In this view, crosstalk with the MAPK signaling might be worth further investigate as well.

3.- In fig 2., authors imaged pancreatic tumor tissue from KPF mouse model combined with Fsp-Cre. The TDTomato reporter allowed to visualize the population of cells targeted by Fsp-Cre during tumorigenesis. Notably, the proportion of Tomato+ cells is

significantly lower compared to aSMA+ cells. What would be the conclusions from this observation?

4.- Results in fig 4 revealed 32% decrease in overall collagens in STAT3 KO tissue samples pointing to a role of STAT3 in ECM remodeling. To address whether this difference in ECM was due to reduced number or reduced activity of CAFs in the STAT3 KO cohort, authors conducted immunofluorescence against aSMA which showed a 36% decrease in total aSMA+ cells. Nonetheless, the possibility of genetic dependencies between STAT3 and aSMA need to be clarified. If this is the case, it might be worth checking on other myCAF markers such as TIMP3 or COL12A1.

5.- Remarkably, loss of STAT3 in CAFs KO didn't have a significant effect in CAF proliferation/survival since no significant changes in Ki67 or cleaved-Caspase-3 were observed. Was that expected given the role of STAT3 in cell growth, differentiation, and survival? What mechanisms could be blocking activation of the apoptotic response in this context? In this view, it might be worth considering a potential STAT3-dependent c-FLIP upregulation as a major resistance mechanism to apoptosis.

6.- Authors hypothesize that STAT3 signaling in CAFs is involved only in macrophage M2-like polarization since recruitment of F480+ macrophags within the TME was not observed (fig 6). As stated by the authors, it is striking that despite the reduction in total aSMA+ cells recruitment of macrophags was not enhanced. In this respect, how the modulation of aSMA+ cells (myCAF) might be affecting other CAF subtypes might be worth exploring since other CAF subtypes (i.e. iCAF) could prevent macrophage and T cell recruitment within the TME.

7.- Loss of CAF-STAT3+ cells led to an increase in T cells and decrease Treg in tumor lesions, which may be a consequence of the decreased number of M2. Alternatively, it may indicate direct communication between CAF and T-cell subsets. Is there any evidence in the literature supporting this idea?

8.- What does the PTEN/STAT3 expression of fibroblasts/cells surrounding normal pancreatic ducts look like? Is this a function gained during activation? At some point, the claim is made that there is a loss of PTEN in cells of the stroma in KPC mouse - is it a loss or simply an absence? Disambiguate or compare as appropriate.

Specific comments:

The high quality of the study is remarkable, and the manuscript is well-written. In any case, I would suggest being more concise to avoid ambiguity in some paragraphs. In example, in pp 11-14, CAFs STAT3 are mentioned several times. It would be good to refer to them as CAF-expressing STAT3 or CAF-STAT3+ cells.

Figure 5b is not referred to in the text.

Supplementary figures 1b-e, 2a are not referred to in the text.

Figures 1b,c, and 6b are mentioned out of order in the text. Author guidelines indicate that figures "should" follow the text. Given that this is a "should" rather than a "must" I recommend that the figures should remain in their current order since it aids the logical flow (in the case of figure 1a,b,c) and presentation (in the case of figure 6b).

Figure 2b y-axis label and/or figure legend could be clarified to indicate that this refers to the percentage of cell number rather than, for instance, intensity.

Figure 3e graph size is very small; suggest making larger.

Supplementary Figure 2 C - left graph Y-axis seems to have PDGFRB+ twice - which is at odds with the figure legend.

It appears that the figure legend for Supplementary figure 4 a and 4b are the wrong way round.

There are points where aSMA+ and SMA+ are written interchangeably; suggest picking one for consistency.

There are points where F480 and F4/80 are written interchangeably; suggest picking one for consistency.

Fig. 1G. Authors claim that p-stat3 is increased when PTEN is knocked down, there is an indication that this is the case, but only one of the two primary PDAC patient-derived CAF cell displayed significant differences. Text should be updated to be less definite.

Fig. 4 Authors show a difference in immune cell infiltration between STAT3wt and STAT3cKO. To further strengthen the difference in immunosuppression consider including quantification of the Ratio between CD3+/CD8+ and CD4FOXP3+ cells in each of the STAT3wt and STAT3cKO tumors, as a greater ratio (high CD3C8+ low CD4FOXP3+) indicate a greater anti-tumor immune activity.

Reviewer #2 (Comments to the Authors (Required)):

Lefler et al. demonstrated Fsp1-Cre - driven depletion of Stat3 led to reduced PDAC progression in autologous KPF pancreatic tumor model. The authors attributed this observation to CAF-secreted CXCL1-mediated phenotypic change of macrophage. The conclusion is questionable due to these two reasons:

1. The specificity of Fsp1-Cre is in question. In contrast to its name, fibroblast specific protein 1, Fsp1 is not a fibroblast-specific marker. Expression of Fsp1, also known as S100a4, is detected in inflammatory macrophages (the M2-like macrophages). The authors should exclude the possibility that Fsp1-Cre-driven Stat3 conditional KO targets the immune cells, in particular the macrophages. This can be achieved by co-staining CD45 and/or F4/80 with Tomato in PDAC and PanIN lesion collected from KPT-TF mice.

2. aSMA has been known as a marker for a subpopulation of CAFs, the myofibroblasts. This study has used aSMA (and in one supplemental figure PDGFRB) as a pan-CAF marker, which is incorrect (aSMA and PDGFRB are also markers for pericytes). More importantly, as the authors mentioned, Biffi et al. 2018 and others have shown that pSTAT3 is a marker for inflammatory CAFs, and pSTAT3 expression is reduced in aSMA+ myofibroblasts. Therefore, only assessing the aSMA+ population after STAT3 conditional KO overlooked the heterogeneity of CAFs and subsequently invalidated the conclusion. Furthermore, in this study, reduced aSMA+ cells were detected in STAT3 cKO host. It is unclear whether the entire CAF population was reduced in the STAT3 cKO hosts or only the proportion of aSMA+ subpopulation was reduced. The conclusion should be reevaluated after staining PDAC with pan-CAF marker such as PDPN and PDGFRA.

Other point:

The STAT3 WB in Figure 1 F and G look identical to the STAT3 WB in supplemental figure B and D, respectively.

Reviewer #3 (Comments to the Authors (Required)):

The manuscript from Lefler et al. examines the role of cancer-associated fibroblast (CAF)-expressed STAT3 in regulating the tumor microenvironment of pancreatic cancer. The results extends prior studies by the group that identified that PTEN loss in CAFs and reduced overall survival in PDAC, further solidifying an import role for CAFs in the aggressive nature of PDAC. The current studies identify STAT3 activity in CAFs (downstream of PTEN) as an important determinant of PDAC development, at least in part, through TME remodeling and immune suppression. CAF-intrinsic STAT3 activity within the TME, correlating with tumor progression and immune suppression, has been reported across other models. However, the results presented here in a PDAC GEM model rather than transplanted tumors, and the high quality of the experimental design, data, and conclusions still add substantially to what is known about the role of STAT3 in CAFs. Overall, the data clearly distinguish that eliminating STAT3 (from ~40%) of the CAFs result in fairly robust change in tumor size and TME parameters (decreased fibrosis; a doubling of CD3+ and CD8+ T-cells, a decrease in T-Reg cells, and a decrease in the proportion of macrophages with an M2 phenotype and increase in M1). The study is particularly relevant for PDAC, as it is a highly stromal tumor, which has important implications for therapeutic effectiveness, including chemotherapies and immune-activating therapies. Below are some concerns that should be addressed prior to publication.

Major concern:

Are data from Figures 4, 5, 6, and 7 from tumors isolated at the same timepoints ("~90-100 days") when the tumors can be at different sizes? If so, would the Stat3-KO tumors look comparable to WT if given sufficient time (i.e. if a 0.5 g KO tumor was compared to a 0.5 g WT tumor)? There should be some clarity about how the tumors that were examined were chosen. For example, while the M&M states that "at least three multispectral images per animal with at least three mice per genotype were acquired", it is not clear how tumors were selected for each analysis. This has implications for the overall conclusion of the paper, since clearly STAT3 KO slows tumor growth and CAF density. However, if the other resulting outcomes occur because of reduced tumor growth or fibroblast density (if they were measured in different sized tumors), then they may be indirect and distinct from the in vitro data showing that STAT3 in CAFs controls macrophage-modulating molecules. Discussions of each possibility should be included so that the reader is clearly aware of what tumor sizes were examined.

Minor concerns:

Revisit the final sentence of the abstract: "The results provide a potential mechanism by which CAFs promote an immune-suppressive TME and "limit tumor progression" in a spontaneous model of PDAC." It would seem one should conclude that CAFs promote, rather than limit tumor progression.

In Figure 1, the authors state "that PTEN protein expression was absent in approximately 80% of SMA+ cells located in the tumor microenvironment" and "We observed a similar loss of PTEN expression in the stroma in the KPC mouse model". Is PTEN expressed at a higher level in normal pancreas fibroblasts, and then lost in CAFs? Is that what is meant by loss of expression in the stroma? If so, then data should be shown or cited.

There is a loss of SMA+ cells (reduction by 36% discussed), but no reduction of KI67+ CAFs and no increase in apoptosis. Why is the number of SMA+ CAFs reduced? This should be reconciled in the discussion.

Supplementary Figure 4. The yellow dots representing double-positive cells is quite difficult to discern. Using another color would be helpful.

Supplementary Fig. 5. The methods are confusing and should be clarified with additional detail. Are these fibroblasts isolated from KPF mouse prior to tumor formation or after tumor formation?

One of the final sentences in the discussion, "Our study indicates that JAK/STAT inhibition could work in two distinct ways: 1) decrease the fibrotic stroma and 2) limit immunosuppressive pathways" should be re-visited, as inhibiting "JAK/STAT" generically would also inhibit type-I interferon responses (JAK/STAT1/2), which are generally considered immune-activating and important for the efficacy of many chemotherapies.

Reviewer Figure 1. Dual aSMA staining on KPF-TF pancreas sections.

Reviewer #1:

1.- Findings pointed to an upregulation of STAT3 in SMA+ CAF cells; nonetheless, the role of IL6-STAT3 axis in the modulation of different CAF populations (i.e. myCAF and iCAF) still needs clarification. As stated by the authors, the identification of CAFs subtypes within the PDAC TME with both tumor-promoting and tumor-limiting characteristics emphasizes the need of a better understanding of the CAF contribution to the PDAC pathogenesis.

Therefore, I would point out that the problem of CAF subtype identity could be better addressed. The authors are aware of the myCAF and iCAF subtypes, for example, however their use of aSMA as a CAF marker more specifically examines myCAFs rather than iCAFs. In this study, then, associations between markers and aSMA expression are examining their association with myCAFs rather than any other. This does not invalidate the findings of the study, and indeed the correlation of those findings with PDGFRb is good (since there are PDGFRb-expressing iCAFs and myCAFs). It would be good to see addressed this limitation of using aSMA and how this may alter

the findings in the study. For example, the effects of STAT3 ablation on collagen deposition may be more relevant to myCAFs, whereas iCAFs are associated with secretion of inflammatory cytokines. Does STAT3 have different roles in different CAF subtypes?

Reviewer #2:

2. aSMA has been known as a marker for a subpopulation of CAFs, the myofibroblasts. This study has used aSMA (and in one supplemental figure PDGFRB) as a pan-CAF marker, which is incorrect (aSMA and PDGFRB are also markers for pericytes). More importantly, as the authors mentioned, Biffi et al. 2018 and others have shown that pSTAT3 is a marker for inflammatory CAFs, and pSTAT3 expression is reduced in aSMA+ myofibroblasts. Therefore, only assessing the aSMA+ population after STAT3 conditional KO overlooked the heterogeneity of CAFs and subsequently invalidated the conclusion. Furthermore, in this study, reduced aSMA+ cells were detected in STAT3 cKO host. It is unclear whether the entire CAF population was reduced in the STAT3 cKO hosts or only the proportion of aSMA+ subpopulation was reduced. The conclusion should be reevaluated after staining PDAC with pan-CAF marker such as PDPN and PDGFRA.

Response: Given the similar questions from Reviewer #1 and Reviewer #2, we will address both in the response that follows. The reviewers make an excellent point, and indeed, the identification of CAF subtypes is an important ongoing line of investigation. However, the use of aSMA in iCAF/myCAF classification is not binary, that is, expression of aSMA and other fibroblast markers are not absent in iCAFs, but instead are significantly lower than in myCAFs. To illustrate this, we invite the reviewer to see below an example of how aSMA staining captures both high and low expressing CAFs (Reviewer Figure 1). This staining also highlights another difference between myCAFs and iCAFs, that the former are located proximal to the tumor, while the latter are more distal. We would like to point out to the Reviewers that in the original paper describing the iCAF/myCAF phenotype (Ohlund et al. 2017, PMID: [28232471](https://pubmed.ncbi.nlm.nih.gov/28232471/)), the authors used aSMA to mark both the iCAFs and myCAFs. Thus, both iCAF and myCAF populations express aSMA.

In regards to the use of PDGFRB, we are aware that this protein can mark pericytes/perivascular cells, similar to aSMA. However, these perivascular cells were easily excluded from our analysis given they are quite distinctive surrounding blood vessels in the sections (see Reviewer Figure 1). Further, we argue that PDGFRB is a perfectly reasonable marker to use to characterize total fibroblast populations in our sections based on previously published data. In the Ohlund et al. 2017 paper, while they used PDGFRA to isolate CAFs from KPC mouse tumors, qPCR analysis demonstrated that these CAFs had the same expression levels of PDGFRB and Acta2 (see Figure 4D in Ohlund et al. 2017). Further, Ohlund et al. use PDGFRB as another fibroblast marker, to co-stain with IL-6 to characterize iCAFs in their model (see Figure 4F,G in Ohlund et al. 2017). What could be appreciated by this co-staining strategy, is that not all PDGFRB+ cells were also positive for the iCAF marker IL-6.

We have additionally supported our view of these markers using the published single cell RNA sequencing data from the Elyada et al. 2019 paper. Using the defined markers of iCAFs and myCAFs for their enriched fibroblast population, one can appreciate there is *Acta2* (aSMA) expression in the defined iCAF cluster, just not as high as in myCAFs (see Reviewer Figure 2). Similarly, *Pdgfrb* expression is present in both subsets. To perform this analysis, we took the fibroblast enriched fraction and analyzed it using Seurat v.4.0. Briefly, cells that had less than 25% of mitochondrial gene expression, with minimum of 300 and maximum of 7500 features detected were selected for the analysis. Cell cycle genes were regressed using SCT transformation. Based on clustree algorithm, we used a resolution of 0.3 which distinctly clustered iCAFs and myCAFs. The fibroblast clusters were subsetted and queried for the expression of *Pdgfrb* and *Acta2* (aSMA).

Importantly, Ohlund et al. point out that the iCAF phenotype can be transient. When iCAFs are plated in monolayer, they can be reverted back to a myCAF state with higher aSMA expression, suggesting “CAFs are dynamic and can assume different phenotypes based on their spatial and biochemical niche within the PDA microenvironment” (Ohlund et al. 2017). Therefore, CAF subtypes are likely constantly changing within the microenvironment and our immunofluorescent staining is providing a snapshot of those dynamics.

The reviewer makes an intriguing point that perhaps STAT3 could play a different role in different subtypes, i.e., play a direct role in collagen deposition in myCAFs and concurrently be more responsible for the inflammatory phenotype in iCAFs. The reviewer points to the idea that collagen deposition may be more relevant to myCAFs, however, iCAFs also express *Col1* genes (albeit at lower levels than myCAF) and high levels of genes such as *Col14a1* and *Has*, which also play a role in ECM dynamics (Elyada et al. 2019). Therefore, both subtypes could influence collagen deposition, and STAT3 could affect this in both populations. This points to the complex nature of trying to explain activities of CAFs within the stroma based on a simple, binary categorization. We don't want to minimize the importance of understanding and characterizing CAF subtypes within the TME, but we feel it is beyond the scope of our current study.

Reviewer Figure 2. Clustering of CAFs from fibroblast enriched data-set from Elyada et al. 2019.

Red dotted line: iCAF cluster; Green dotted line myCAF cluster.

Specific Comments for Reviewer #1:

2.- To further confirm the relationship between PTEN and activated STAT3, authors have assessed whether P-STAT3 was increased in PTEN KO fibroblasts. PTEN is a known negative regulator of STAT3 which depletion in CAFs induced increased P-STAT3 as expected (fig1). Further mechanistic studies are needed to elucidate whether the effect comes from the canonical RTK/JAK or non-canonical PI3K/TEC pathways as stated by the authors. In this view, crosstalk with the MAPK signaling might be worth further investigate as well.

Response: We have added these experiments to the main body of the paper in Figure 1 and in Supplemental data. In order to elucidate whether the increase in STAT3 activation originates from canonical RTK/JAK signaling or non-canonical PI3K pathway, we treated WT and PTEN null fibroblasts with a JAK inhibitor (Momelotinib), and PI3K inhibitors (Supplemental Fig. 1F). We found that treatment with JAKi results in a decrease in STAT3 phosphorylation in both PTEN WT and PTEN null CAFs (new manuscript Figure 1H, I In contrast, the pan-PI3 kinase inhibitor and the isoform-specific inhibitors did not have a significant effect on STAT3 phosphorylation (Supplemental Fig. 1F). Therefore, we conclude from these in vitro experiments that in PTEN null pancreatic tumor fibroblasts, constitutive STAT3 phosphorylation is driven by the JAK pathway. We speculate that PTEN loss leads to direct activation of RTK signaling or upregulation of growth factors/cytokines like IL6 that act in an autocrine fashion to stimulate STAT3 phosphorylation.

3.- In fig 2,, authors imaged pancreatic tumor tissue from KPF mouse model combined with Fsp-Cre. The TDTomato reporter allowed to visualize the population of cells targeted by Fsp-Cre during tumorigenesis. Notably, the proportion of Tomato+ cells is significantly lower compared to aSMA+ cells. What would be the conclusions from this observation?

Response: Our conclusions from this observation are that not all aSMA+ cells are targeted by the FspCre. We made a point to emphasize this in our discussion, stating that we are indeed targeting a small population of CAFs in this study.

4.- Results in fig 4 revealed 32% decrease in overall collagens in STAT3 KO tissue samples pointing to a role of STAT3 in ECM remodeling. To address whether this difference in ECM was due to reduced number or reduced activity of CAFs in the STAT3 KO cohort, authors conducted immunofluorescence against aSMA which showed a 36% decrease in total aSMA+ cells. Nonetheless, the possibility of genetic dependencies between STAT3 and aSMA need to be clarified. If this is the case, it might be worth checking on other myCAF markers such as TIMP3 or COL12A1.

Response: See our response to comment #1.

5.- Remarkably, loss of STAT3 in CAFs KO didn't have a significant effect in CAF

proliferation/survival since no significant changes in Ki67 or cleaved-Caspase-3 were observed. Was that expected given the role of STAT3 in cell growth, differentiation, and survival? What mechanisms could be blocking activation of the apoptotic response in this context? In this view, it might be worth considering a potential STAT3-dependent c-FLIP upregulation as a major resistance mechanism to apoptosis.

Response: Although our hypothesis was that we might see a reduction in Ki-67 and an increase in CC3, a major limitation of ki67 and CC3 staining in our sections only reveals a snapshot of the dynamic process of cellular proliferation and apoptosis that is actually occurring *in vivo*. However, other mechanisms could be acting independently of STAT3 to promote cell survival and proliferation pathways such as Src and Pi3K signaling. Pi3k signaling is a strong activator of cyclin D, an important player in cell cycle progression. Another interesting pathway this reviewer brings up is c-FLIP, as STAT3 has been shown to transcriptionally silence c-FLIP expression. Therefore, this could be an interesting mechanism to investigate. Exploring potential mechanisms by which these CAFs block the activation of the apoptotic response is intriguing, but beyond the scope of this study.

6.- Authors hypothesize that STAT3 signaling in CAFs is involved only in macrophage M2-like polarization since recruitment of F480+ macrophages within the TME was not observed (fig 6). As stated by the authors, it is striking that despite the reduction in total aSMA+ cells recruitment of macrophages was not enhanced. In this respect, how the modulation of aSMA+ cells (myCAF) might be affecting other CAF subtypes might be worth exploring since other CAF subtypes (i.e. iCAF) could prevent macrophage and T cell recruitment within the TME.

Response: This is an intriguing question; however we believe we have addressed this in response to comment #1.

7.- Loss of CAF-STAT3+ cells led to an increase in T cells and decrease Treg in tumor lesions, which may be a consequence of the decreased number of M2. Alternatively, it may indicate direct communication between CAF and T-cell subsets. Is there any evidence in the literature supporting this idea?

Response: We have added a comment on this in our discussion. Though priming of the T cell response relies heavily upon macrophages and dendritic cells, there have been some studies that indicate CAFs could directly affect T cell activity in the PDAC TME. Pancreatic CAFs do express immune checkpoints such as PD-L1 and PD-L2. Investigators have shown that blockade of PL-1 can restore CAF-mediated T cell suppression however further studies are needed to evaluate direct interactions between CAFs and T cells (Nazareth MR et al. 2007, PMID: 17442937). CAFs can also disrupt T cell functionality through the secretion of soluble factors such as COX-2, PGE₂ and TGF-B (reviewed in Gorchs and Kaipe 2021, PMID: 34203869). Though we did already address the possibility that our

result was from direct interaction between CAF and T cell, we have added more details on the current literature in our discussion.

8.- What does the PTEN/STAT3 expression of fibroblasts/cells surrounding normal pancreatic ducts look like? Is this a function gained during activation? At some point, the claim is made that there is a loss of PTEN in cells of the stroma in KPC mouse - is it a loss or simply an absence? Disambiguate or compare as appropriate.

Response: We have clarified that what we meant, is there is an absence of PTEN protein expression in the CAFs in the stroma in the KPC mouse (more precise language).

Figure 5b is not referred to in the text.

-Corrected

Supplementary figures 1b-e, 2a are not referred to in the text.

- Corrected

Figures 1b,c, and 6b are mentioned out of order in the text. Author guidelines indicate that figures "should" follow the text. Given that this is a "should" rather than a "must" I recommend that the figures should remain in their current order since it aids the logical flow (in the case of figure 1a,b,c) and presentation (in the case of figure 6b).

Figure 2b y-axis label and/or figure legend could be clarified to indicate that this refers to the percentage of cell number rather than, for instance, intensity.

- Corrected

Figure 3e graph size is very small; suggest making larger.

- Corrected

Supplementary Figure 2 C - left graph Y-axis seems to have PDGFRB+ twice - which is at odds with the figure legend.

- Corrected

It appears that the figure legend for Supplementary figure 4 a and 4b are the wrong way round.

-Corrected

There are points where aSMA+ and SMA+ are written interchangeably; suggest picking one for consistency.

-Corrected

There are points where F480 and F4/80 are written interchangeably; suggest picking one for consistency.

-Corrected

Fig. 1G. Authors claim that p-stat3 is increased when PTEN is knocked down, there is an indication that this is the case, but only one of the two primary PDAC patient-derived CAF cell displayed significant differences. Text should be updated to be less definite.

-Corrected

Fig. 4 Authors show a difference in immune cell infiltration between STAT3wt and STAT3cKO. To further strengthen the difference in immunosuppression consider including quantification of the Ratio between CD3+/CD8+ and CD4FOXP3+ cells in each of the STAT3wt and STAT3cKO tumors, as a greater ratio (high CD3C8+ low CD4FOXP3+) indicate a greater anti-tumor immune activity.

-DONE

Specific Comments for Reviewer #2:

1. The specificity of Fsp1-Cre is in question. In contrast to its name, fibroblast specific protein 1, Fsp1 is not a fibroblast-specific marker. Expression of Fsp1, also known as S100a4, is detected in inflammatory macrophages (the M2-like macrophages). The authors should exclude the possibility that Fsp1-Cre-driven Stat3 conditional KO targets the immune cells, in particular the macrophages. This can be achieved by co-staining CD45 and/or F4/80 with Tomato in PDAC and PanIN lesion collected from KPT-TF mice.

Response: It is important to make clear that the Fsp-Cre transgene used here was constructed and validated by our lab, and to note that the expression of the transgene does not recapitulate expression of the endogenous FSP1/S100A4 gene. Our Fsp-Cre is distinct from the Fsp-Cre available through Jackson labs, which was originally constructed by Eric Nielson's lab at Vanderbilt. The Jackson Fsp-Cre can indeed target both fibroblasts and macrophages (among other cell compartments). We have published an extensive analysis of the Fsp-Cre developed by our group to demonstrate its specificity in the mammary gland (Trimboli et al. 2008, PMID: 18245497. See Figure 2B and 2C; Trimboli et al. 2009, PMID: 19847259 see Figure 1a and Supplementary Figure 2a and b, which demonstrate there is no expression of the transgene in cytokeratin-positive cells, F480 macrophages, and CD31-endothelial cells). In pancreatic cancer models, we have also shown the specificity. (see Liu et al. 2016, PMID: 27633013 see Figure 1a and supplemental Figure 1b). To demonstrate again the specificity of the Fsp-Cre transgene in the current experiments, we have added an additional supplemental figure that shows that co-staining of F480 and tomato in the KPF-TF model. Using the same quantification methods for our other IF staining, we can demonstrate that there are extremely low levels of colocalization between F480 and tomato, an average of 0.07% double positive F480 and tomato cells per multi-spectral stamp.

Other point:

The STAT3 WB in Figure 1 F and G look identical to the STAT3 WB in supplemental figure B and D, respectively.

Response: The STAT3 WB in Figure 1F and G are identical to the STAT3 WB in supplemental Figure B and D. This was because total STAT3 was measured on a separate blot made using a gel loaded with the same protein extract to avoid potential problems with stripping and re-probing the first blot. Supplemental Figure B and D exhibit the second blot that was run alongside the pSTAT3 blots shown in the main figure. We clarified this in the figure legends of the figures.

Specific Comments for Reviewer #3:

Are data from Figures 4, 5, 6, and 7 from tumors isolated at the same timepoints ("~90-100 days") when the tumors can be at different sizes? If so, would the Stat3-KO tumors look comparable to WT if given sufficient time (i.e. if a 0.5 g KO tumor was compared to a 0.5 g WT tumor)? There should be some clarity about how the tumors that were examined were chosen. For example, while the M&M states that "at least three multispectral images per animal with at least three mice per genotype were acquired", it is not clear how tumors were selected for each analysis. This has implications for the overall conclusion of the paper, since clearly STAT3 KO slows tumor growth and CAF density. However, if the other resulting outcomes occur because of reduced tumor growth or fibroblast density (if they were measured in different sized tumors), then they may be indirect and distinct from the in vitro data showing that STAT3 in CAFs controls macrophage-modulating molecules. Discussions of each possibility should be included so that the reader is clearly aware of what tumor sizes were examined.

Response: Our analyses of STAT3^{WT} vs STAT3^{CKO} pancreata were done between regions carefully selected to have similar histopathology. Using the Vectra Polaris whole slide scanner, we were able to carefully stamp out regions that were similar in histopathology between the two cohorts to control for overall differences in tumor progression. Thus we compared regions with late PAN-In and early invasive carcinoma, and excluded advanced tumors and necrotic regions.

Minor concerns:

Revisit the final sentence of the abstract: "The results provide a potential mechanism by which CAFs promote an immune-suppressive TME and "limit tumor progression" in a spontaneous model of PDAC." It would seem one should conclude that CAFs promote, rather than limit tumor progression.

-We revised this sentence as suggested.

In Figure 1, the authors state "that PTEN protein expression was absent in approximately 80% of SMA+ cells located in the tumor microenvironment" and "We observed a similar loss of PTEN expression in the stroma in the KPC mouse model". Is

PTEN expressed at a higher level in normal pancreas fibroblasts, and then lost in CAFs? Is that what is meant by loss of expression in the stroma? If so, then data should be shown or cited.

Response: This language has been addressed. What we meant, is that there is an overall absence of PTEN expression in the stroma in the KPC mouse model.

There is a loss of SMA+ cells (reduction by 36% discussed), but no reduction of KI67+ CAFs and no increase in apoptosis. Why is the number of SMA+ CAFs reduced? This should be reconciled in the discussion.

Response: see response to comment #5 posed by reviewer #1.

Supplementary Figure 4. The yellow dots representing double-positive cells is quite difficult to discern. Using another color would be helpful.

Response: I changed the yellow to a green that we think is easier to discern.

Supplementary Fig. 5. The methods are confusing and should be clarified with additional detail. Are these fibroblasts isolated from KPF mouse prior to tumor formation or after tumor formation?

Response: We have clarified this in the methods and the figure legend. These were fibroblasts isolated from pancreas tumors from the KPF mice.

One of the final sentences in the discussion, "Our study indicates that JAK/STAT inhibition could work in two distinct ways: 1) decrease the fibrotic stroma and 2) limit immunosuppressive pathways" should be re-visited, as inhibiting "JAK/STAT" generically would also inhibit type-I interferon responses (JAK/STAT1/2), which are generally considered immune-activating and important for the efficacy of many chemotherapies.

Response: We should have been clearer in our language and have changed our sentence to say "inhibition of STAT3 signaling".

June 17, 2022

Re: Life Science Alliance manuscript #LSA-2022-01460-TR

Dr. Michael C Ostrowski
Medical University of South Carolina
Hollings Cancer Center
86 Jonathon Lucas Street
HO 512D
Charleston, SC 29425

Dear Dr. Ostrowski,

Thank you for submitting your revised manuscript entitled "STAT3 in Tumor Fibroblasts Promotes an Immunosuppressive Microenvironment in Pancreatic Cancer" to Life Science Alliance. The manuscript has been seen by the original reviewers whose comments are appended below. While the reviewers continue to be overall positive about the work in terms of its suitability for Life Science Alliance, some important issues remain.

Our general policy is that papers are considered through only one revision cycle; however, given that the suggested changes are relatively minor, we are open to one additional short round of revision. Please note that I will expect to make a final decision without additional reviewer input upon re-submission.

Please address Reviewer 2's remaining comments. Please also address Reviewer 1's question about the distribution of tomato-positive CAFs. The other points made by Reviewer 1 do not need to be addressed experimentally, unless that data is readily available.

Please submit the final revision within one month, along with a letter that includes a point by point response to the remaining reviewer comments.

To upload the revised version of your manuscript, please log in to your account: <https://lsa.msubmit.net/cgi-bin/main.plex>
You will be guided to complete the submission of your revised manuscript and to fill in all necessary information.

- A letter addressing the reviewers' comments point by point.
- An editable version of the final text (.DOC or .DOCX) is needed for copyediting (no PDFs).
- High-resolution figure, supplementary figure and video files uploaded as individual files: See our detailed guidelines for preparing your production-ready images, <https://www.life-science-alliance.org/authors>
- Summary blurb (enter in submission system): A short text summarizing in a single sentence the study (max. 200 characters including spaces). This text is used in conjunction with the titles of papers, hence should be informative and complementary to the title and running title. It should describe the context and significance of the findings for a general readership; it should be written in the present tense and refer to the work in the third person. Author names should not be mentioned.

B. MANUSCRIPT ORGANIZATION AND FORMATTING:

Sincerely,

Reviewer #1 (Comments to the Authors (Required)):

Authors have now addressed some relevant questions such as the role of the canonical RTK/JAK pathway in the activation of STAT3 and have offered reasonable arguments regarding the identity of aSMA+ cells. In addition, authors have carefully rectified any ambiguous statements from the original manuscript and have included additional data further supporting their conclusions that loss of the tumor suppressor PTEN in pancreatic CAFs lead to an enhancement of the STAT3 signalling and secretion of CXCL1 which favoured M2 polarization.

The identification of CAF subtypes is certainly an important and rather complex question due to the heterogeneity and transient nature of CAF subtypes; and therefore, we understand that the authors think that this matter might be beyond the scope of the current study, but I think it still needs to be address to be able to fully validate the conclusions made. Reviewer figure 1 nicely illustrates the spatial distribution difference in expression levels of aSMA in CAFs, where myCAFs are aSMA high and positioned juxtapost to cancer cells, while iCAFs are found deeper in the stroma and aSMA low. Therefore, it would be possible for the authors to quantify PTEN and pSTAT3 and taking the subtypes (spatial localization and staining intensity) into consideration. This is important since it is also stated in the conclusion that STAT3 activation is characteristic for the iCAFs. Is pSTAT3 more frequent seen in iCAFs in the model used?

Also, in this study, only a small fraction of CAFs are activating the FSP-cre construct and become tomato-positive. This can indicate that only a specific subtype of CAFs is activating cre, and that would mean that the effect seen on CXCL1 secretion and M2 polarization is driven by this specific subtype, and not by CAFs in general.

But it can also be that cre activation is just randomly distributed between all different CAF subtypes. By analysing how the tomato-positive CAFs are distributed this question could be addressed.

Reviewer #2 (Comments to the Authors (Required)):

The authors have addressed my concern by co-staining F480 and Tomato to clarify the specificity of their Fsp-cre mouse strain. Although the lack of co-localization (Fig S3C) is not very meaningful in my opinion, as F480 should express on the membrane and Tomato should be in the cytoplasm, the low number of double positive cells (Fig S3D) seem to demonstrate that their transgene-based Fsp-cre strain doesn't label macrophages. However, as the expression of Fsp is not fibroblast specific, the author should include the reasoning of performing F480/Tomato co-stain in the text and cite relevant article(s) (for example - Fibroblast-specific protein 1 identifies an inflammatory subpopulation of macrophages in the liver. Österreicher CH, Penz-Österreicher M, Grivennikov SI, Guma M, Koltsova EK, Datz C, Sasik R, Hardiman G, Karin M, Brenner DA. Proc Natl Acad Sci U S A. 2011 Jan 4;108(1):308-13.) to keep the readers well informed.

The authors' responses regarding the usage of aSMA+ and PDGFRB as CAF markers are reasonable. I strongly encourage the authors to include Reviewer Figure 1 as a supplemental panel to inform the readers how they have used aSMA (aka. exclude the pericytes and include the low aSMA cells) as a panCAF marker in their image analysis.

I agree with the authors that there are shared gene expressions among CAF subtypes and these CAF subtypes are dynamic. However, as demonstrated in figure 2C, the authors have the capacity to assess whether the Fsp-cre mediated STAT3 cKO are predominately targeting the aSMA^{low} CAFs. The gained knowledge will better link this study to current literature.

Lastly, I disagree with the authors on how to display the STAT3 WB. Since the STAT3 WB is from a separately loaded gel, it should be displayed with its own loading control as a separated blot. Including STAT3 lane from a separated blot is misleading. The author should remove the STAT3 lane from Figure 1 F and G.

Reviewer #3 (Comments to the Authors (Required)):

The authors have addressed the concerns noted during the original review. There are no further concerns.

We appreciate the opportunity to address these last minor reviewer concerns. We have made the changes requested and hope that our manuscript now meets editorial standards for publication in *Life Science Alliance*. Point-by-point responses to the remaining questions follow.

Reviewer #1: But it can also be that cre activation is just randomly distributed between all different CAF subtypes. By analysing how the tomato-positive CAFs are distributed this question could be addressed.

Reviewer #2: However, as demonstrated in figure 2C, the authors have the capacity to assess whether the Fsp-cre mediated STAT3 cKO are predominately targeting the aSMA^{low} CAFs. The gained knowledge will better link this study to current literature.

Quantification of images (n=4 mice, 4 images per mouse) demonstrates 32% Tomato+SMA^{hi} vs. 68% Tomato+SMA^{lo} cells in the total RFP+SMA+ population (Figure 2E). This result addresses both reviewers questions and is consistent with Fsp-Cre activity in a subset of iCAFs and myCAFs. We've modified text in Results, and in Discussion to include the implications of this finding.

Reviewer #2 (Comments to the Authors (Required)):

The authors have addressed my concern by co-staining F480 and Tomato to clarify the specificity of their Fsp-cre mouse strain. Although the lack of co-localization (Fig S3C) is not very meaningful in my opinion, as F480 should express on the membrane and Tomato should be in the cytoplasm, the low number of double positive cells (Fig S3D) seem to demonstrate that their transgene-based Fsp-cre strain doesn't label macrophages. However, as the expression of Fsp is not fibroblast specific, the author should include the reasoning of performing F480/Tomato co-stain in the text and cite relevant article(s) (for example - Fibroblast-specific protein 1 identifies an inflammatory subpopulation of macrophages in the liver. Österreicher CH, Penz-Österreicher M, Grivennikov SI, Guma M, Koltsova EK, Datz C, Sasik R, Hardiman G, Karin M, Brenner DA. Proc Natl Acad Sci U S A. 2011 Jan 4;108(1):308-13.) to keep the readers well informed.

We have now included in the text of the Results section that 1) FSP-/S100A4 is expressed in macrophages (citing this paper), however, as documented in our previous publications our FSP-Cre does not recapitulate the expression of the S100A4 gene probably due the insertion site of the transgene array, and is distinct from the FSP-cre made in Eric Neilson's lab; 2) our previous publications and data in supplemental clearly document that our Fsp-Cre is not expressed in macrophages.

The authors' responses regarding the usage of aSMA+ and PDGFRB as CAF markers are reasonable. I strongly encourage the authors to include Reviewer Figure 1 as a supplemental panel to inform the readers how they have used aSMA (aka. exclude the pericytes and include the low aSMA cells) as a panCAF marker in their image analysis.

We have included this image in Supplemental Figure 3E.

Lastly, I disagree with the authors on how to display the STAT3 WB. Since the STAT3 WB is from a separately loaded gel, it should be displayed with its own loading control as a separated blot. Including STAT3 lane from a separated blot is misleading. The author should remove the STAT3 lane from Figure 1 F and G.

The reviewer's point is well taken, and we have removed the panSTAT3 antibody panel from Figure 1 and now refer to the data in supplemental Fig. 1B&D, which has its own separate protein loading controls.

Thank you for considering our manuscript with these changes, and we await your final decision.

June 24, 2022

RE: Life Science Alliance Manuscript #LSA-2022-01460-TRR

Dr. Michael C Ostrowski
Medical University of South Carolina
Hollings Cancer Center
86 Jonathon Lucas Street
HO 512D
Charleston, SC SC 29425

Dear Dr. Ostrowski,

Thank you for submitting your revised manuscript entitled "STAT3 in Tumor Fibroblasts Promotes an Immunosuppressive Microenvironment in Pancreatic Cancer". We would be happy to publish your paper in Life Science Alliance pending final revisions necessary to meet our formatting guidelines.

-please use the [10 author names, et al.] format in your references (i.e. limit the author names to the first 10)

Figure Check:

-please add sizes next to blots

A. FINAL FILES:

B. MANUSCRIPT ORGANIZATION AND FORMATTING:

Sincerely,

June 30, 2022

RE: Life Science Alliance Manuscript #LSA-2022-01460-TRRR

Dr. Michael C Ostrowski
Medical University of South Carolina
Hollings Cancer Center
86 Jonathon Lucas Street
HO 512D
Charleston, SC SC 29425

Dear Dr. Ostrowski,

Thank you for submitting your Research Article entitled "STAT3 in Tumor Fibroblasts Promotes an Immunosuppressive Microenvironment in Pancreatic Cancer". It is a pleasure to let you know that your manuscript is now accepted for publication in Life Science Alliance. Congratulations on this interesting work.

DISTRIBUTION OF MATERIALS:

Again, congratulations on a very nice paper. I hope you found the review process to be constructive and are pleased with how the manuscript was handled editorially. We look forward to future exciting submissions from your lab.

Sincerely,
